# Trend towards virtual and hybrid conferences may be an effective climate change mitigation strategy

Yanqiu Tao [1], Debbie Steckel[2], Jiří Jaromír Klemeš [3] & Fengqi You [1,4,5 ✉]

Since 2020, the COVID-19 pandemic has urged event holders to shift conferences online. Virtual and hybrid conferences are greener alternatives to in-person conferences, yet their environmental sustainability has not been fully assessed. Considering food, accommodation, preparation, execution, information and communication technology, and transportation, here we report comparative life cycle assessment results of in-person, virtual, and hybrid conferences and consider carbon footprint trade-offs between in-person participation and hybrid conferences. We find that transitioning from in-person to virtual conferencing can substantially reduce the carbon footprint by 94% and energy use by 90%. For the sake of maintaining more than 50% of in-person participation, carefully selected hubs for hybrid conferences have the potential to slash carbon footprint and energy use by two-thirds. Furthermore, switching the dietary type of future conferences to plant-based diets and improving energy efficiencies of the information and communication technology sector can further reduce the carbon footprint of virtual conferences.

[1] Systems Engineering, Cornell University, Ithaca, New York, NY 14853, USA. [2] The American Center for Life Cycle Assessment, Bethesda, MD 20824, USA. [3] Sustainable Process Integration Laboratory – SPIL, NETME Centre, Faculty of Mechanical Engineering, Brno University of Technology - VUT Brno, Technická 2896/2, 616 69 Brno, Czech Republic. [4] Robert Frederick Smith School of Chemical and Biomolecular Engineering, Cornell University, Ithaca, New York, NY 14853, USA. [5] Cornell Atkinson Center for Sustainability, 340 Tower Road Cornell University Ithaca, Ithaca, NY 14853, USA. ✉email: fengqi.you@cornell.edu

The rapid expansion of the global event industry over the last few decades has brought about tourism growth and socio-economic development and also exerted surprisingly large pressure on the goals to mitigate global climate change. In 2017, more than 1.5 billion participants across over 180 countries were involved in business events, contributing to $2.5 trillion of spending and supporting 26 million jobs[1]. In addition, the number of regular international events (≥50 participants) doubles every ten years, and the market size of the trillion-dollar events industry is expected to grow at a rate of 11.2% for this decade[2,3]. As the event industry proliferates, it also leads to substantial greenhouse gas (GHG) emissions. The carbon footprint per participant reaches up to 3000 kg $CO_2$ equivalent as reported by previous life cycle assessment (LCA) studies[4–13], suggesting that the annual carbon footprint for the global event industry are of the same order of magnitude as the yearly GHG emissions of the entire United States (U.S.), responsible for more than 10% of global $CO_2$ emissions[14]. To meet the Paris Agreement target[15], the event organizers adopt sustainability measures[16], and promotion towards the less carbon-intensive virtual conferences has never been discontinued in the past decade[8,17–20]. Since 2020, the COVID-19 pandemic has urged event holders to shift conferences online.

Whether future conferences should return in-person, keep fully virtual, or evolve to a hybrid of both has been widely debated[19]. Common reasons against virtual conferences can be categorized as digital-meeting fatigue, loss of serendipitous hallway conversations, impersonal interactions, and challenging time zones[19]. By contrast, supporters of virtual conferences consider virtual interactions as a far more accessible, inclusive, and sustainable counterpart to the traditional in-person format[18,21–23]. As a compromise of the two, hybrid conferences may be a viable solution[24,25]. However, there is a minimal quantitative understanding of the environmental impacts from different modes of conferences. In order to understand the sustainability implications of future conferences and inform the

policies, it is essential to quantify the environmental footprints of virtual, in-person, and hybrid conferences and also investigate the trade-offs between in-person interactions and the life cycle carbon footprint.

Previous analysis on the travel emissions of in-person conferences and their virtual and multi-hub hybrid counterparts has been conducted[4], but it neglected greenhouse gas emissions from the video-conferencing technology and auxiliary emissions from resource and energy consumptions other than transportation, such as conference planning and preparation, execution, catering, and accommodation during the conference. This simplification may overrate the benefits of virtual conferences and cause discriminating opposition to the in-person conferences. A recent study on the carbon footprint of virtual, in-person, and hybrid conferences accounted for the video-conferencing-related emissions, transportation, execution, catering, and accommodation[26]. However, it considered a single conference hub for both in-person and hybrid conferences and thus neglected the geographical effects of hub selection and participant assignment. Several studies attempted to quantify the environmental impacts from virtual and in-person conferences[5,6], but none of them assessed the environmental sustainability of hybrid conferences. Thus, a critical gap exists in improving the understanding of life cycle environmental impacts of in-person, virtual, and hybrid conferences. In addition, the geographical effects of hub selection and participant assignment have not yet been investigated in the literature, to the best of our knowledge.

To fill these knowledge gaps, the objective of this study is to quantify the life cycle environmental impacts of in-person, virtual, and hybrid conferences and to understand the trade-offs between in-person interactions and the carbon footprint of conferences. Here, we present holistic LCA results on the most concerning carbon footprint and cumulative energy demand (CED) of the in-person, virtual, and hybrid conferences to understand their impacts on climate change, as shown in Fig. 1.

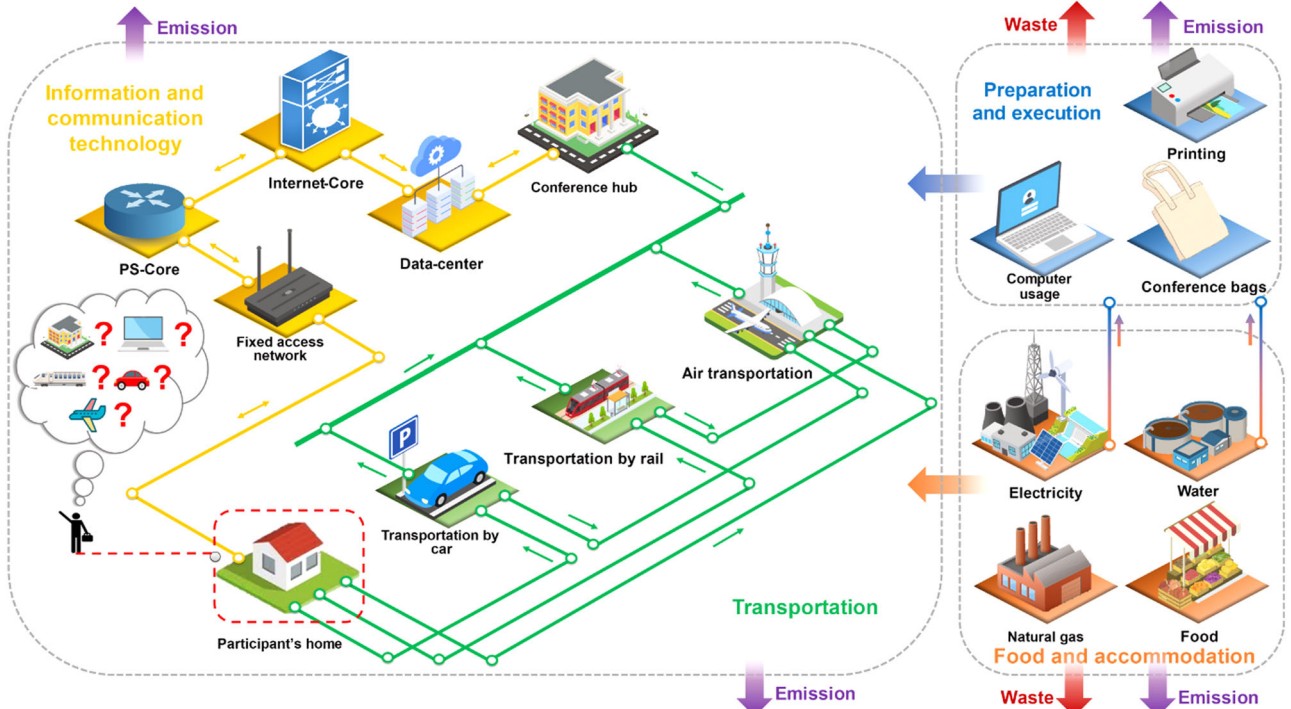

**Fig. 1 Illustration of a hybrid conference that can be accessed by both in-person participants and virtual participants.** Here we demonstrate the hybrid mode of a conference, where virtual participants are connected with in-person participants traveling to the conference hubs through video-conferencing technologies. Yellow lines indicate the virtual pathway, and green lines indicate the in-person pathway to approach the conference.

In addition, a total of 17 ReCiPe midpoint indicators[27], each of which represents a specific aspect of environmental concerns, are adopted to demonstrate the full-spectrum environmental impacts and to identify environmental hotspots. Following the ISO 14040 standard[28], the LCA methodology is constituted by four phases, including goal and scope definition, life cycle inventory (LCI), life cycle impact assessment (LCIA), and interpretation. The four phases of this study are detailed in the Methods section. Hub selections and participant assignments are optimized using a facility location model to assess the potentials of mitigating carbon footprint for in-person conferences with multiple conference hubs. Participants of the 2020 American Center for Life Cycle Assessment virtual conference are used for this case study. We demonstrate that the carbon footprint can be lowered by 94% and CED by 90% when switching from in-person to virtual conferences. Furthermore, we consider two sets of scenarios for the hybrid conferences to investigate the trade-offs between in-person communication and carbon footprint. In particular, the Maximum Virtual Participation (MVP) scenarios introduce a virtual participation level constraint (10%, 30%, 50%, or 70%) into the facility location model to optimize the hub selections and participant assignments for minimizing the travel distance. The Maximum Travel Distance (MTD) scenarios set a maximum travel distance (1 000 km, 3 000 km, 5 000 km, or 10,000 km) for participants who attend in-person while prioritizing in-person participation. The number of hubs ranges from one to six for both sets of scenarios examined. The results show that the MVP scenarios clearly illustrate trade-offs between the carbon footprint and level of in-person participation, while the effects of the MTD scenarios are sensitive to the threshold. Moreover, reducing the carbon footprint and CED of in-person conferences to one-third can be attained with <50% of virtual participation. This study provides insights to the public for understanding the pros and cons of different conference models. The identified environmental hotspots inform the policymakers of the importance of improving energy efficiency and resource utilization in the information and communication technology (ICT) and transportation sectors. Last, this analysis can be adopted to evaluate the environmental impacts of other conferences with the different geographic distribution of participants and to advise the event organizers of hub selections, participant assignments, and desired level of in-person participation, in addition to sustainability measures and concerns on climate change.

## Results

**Comparison with the state-of-the-art conference LCA studies.** Supplementary Table 1 and Supplementary Fig. 1 summarize and compare the state-of-the-art conference LCA studies. Previous LCA studies mainly focused on quantifying the carbon footprint of in-person conferences, while only two of them presented results for other impact categories using the LCIA method, including CML2001, USEtox, Eco-Indicator 99, and UBP 97[6,13]. Half of them focused exclusively on round-trip transportation[4,5,9,11,12,29] while the rest considered life cycle stages of preparation, execution, catering, accommodation, and transportation[6–8,13,26]. However, due to differences in assumptions associated with in-person conferences (e.g., duration, size, and locations of the conference, geographical distribution of participants, transportation mode, system boundary, and selection of characterization factors), the carbon footprint ranges from 92 to 3 540 kg $CO_2$ equivalent per capita. All of these studies identified transportation as the environmental hotspot. The conference site and geographical distribution of participants determine the transportation distance and mode for participants. From those who reported the average transportation distance, the average round-trip transportation distance varies from 1 980 km[13]

to 9 564 km[9]. However, the average distance does not illustrate the complete picture of the participant transportation. It was found that the 10–20% of participants with the most polluting trips contribute to a substantial portion (20–70%) of the total transportation-induced emissions[4–7,12]. These values depend on the distribution of participants, which reveals whether the conference is more localized or more internationalized. As shown in Supplementary Table 1, most participants are from the region where the conferences are held. The conference location is also important in determining the transportation profile. A conference location with better train connection to other major cities is capable of allowing more participants to transport by train and thus has more potential of reducing carbon footprint. In contrast, a conference located in the southern hemisphere usually performs much worse in terms of the carbon footprint than the northern hemisphere[11,12]. Bossdorf et al. suggested food and accommodation accounted for 18% and 13% of the total carbon footprint, respectively[8]. On the other hand, owing to the exclusive vegetarian menu and the much higher transportation emissions, Astudillo and Azarijafari reported that food and accommodation only accounted for 1% and 2% of the total carbon footprint[7].

Recent studies compared the carbon footprint of in-person and virtual conferences[4,5,26], which ranges from 0 to 5.87 kg $CO_2$ equivalent per capita. Specifically, Ewijk and Hoekman assumed carbon neutrality for the virtual conference[4]; Jäckle computed the carbon footprint from the electricity needed for devices and servers[26]; Burtscher et al. considered emissions related to network, laptop, and Zoom-server[5]. Several studies considered multi-site conferences, yet the choices of locations are arbitrary[4,13]. For example, Stroud and Feeley optimized the conference location by minimizing the carbon footprint while restricting the potential conference locations to participants' origins[9]. Astudillo and Azarijafari considered the geometric median of all participants as the optimal conference location[7]. As discussed above, none of the existing studies explicitly explored to what extent virtual conferences and multi-hub hybrid conferences with spatially optimized conference hubs and participant assignments can reduce the environmental impacts of in-person conferences.

**Environmental impacts of the baseline virtual conference.** This "cradle-to-grave" LCA study focuses on life cycle stages of food preparation, accommodation, preparation, execution, information and communication technology, and transportation, as shown in Fig. 1. For a fair comparison, food preparation and accommodation are considered for the virtual, in-person, and hybrid scenarios. Figure 2c presents breakdowns of life cycle carbon footprint, CED, and full-spectrum environmental profiles for the baseline virtual conference. The origins of the virtual participants whose geographic information is available (383 out of 536) are listed in Supplementary Table 2. The resulting carbon footprint and CED are 46 kg $CO_2$ equivalent and 767 MJ equivalent per participant, respectively, equivalent to the GHG emissions and energy consumption of a 150 passenger-kilometer (pkm) road trip in a medium-size petrol car following the Euro 5 emission standards.

Food preparation dominates most impact categories, especially agricultural land occupation, terrestrial ecotoxicity, and water depletion. We adopt the average amount and composition of country-level food supply from FAOSTAT, which is the statistical database of the Food and Agriculture Organization of the United Nations, to represent the food consumption during the conference[30,31]. Major food categories include fruits, vegetables, grain, dairy, protein food, legumes, vegetable oil, animal fats, and sugars. Among the many types of food consumed, beef, milk, and

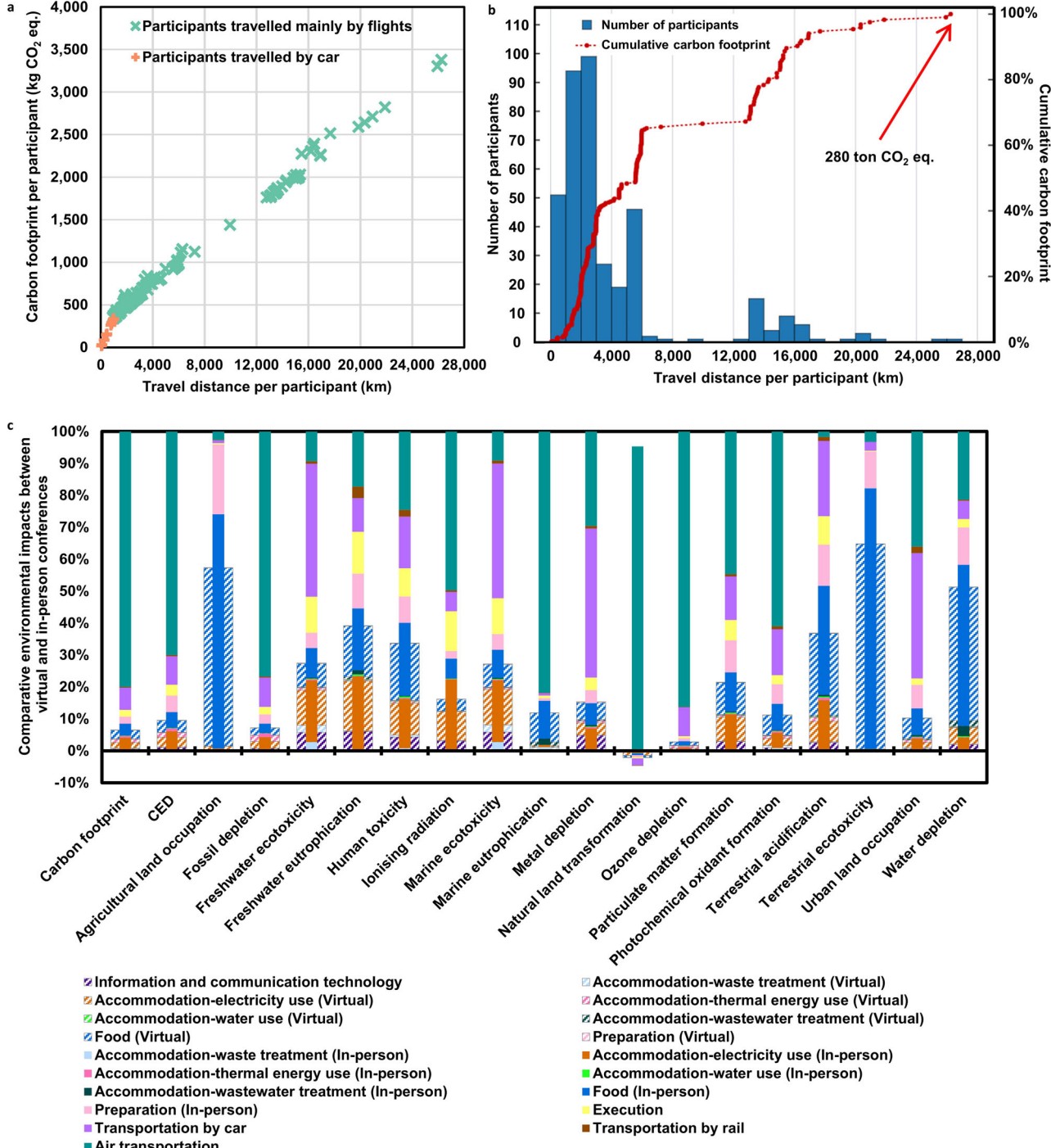

**Fig. 2 Comparison between the 1-hub in-person conference and the virtual conference. a** Carbon footprint associated with transportation for individual participants, indicating that the per pkm carbon footprint for trips primarily by plane (slope of cyan dashed line) tends to be smaller than that for driving (slope of orange dashed line). **b** Cumulative carbon footprint for participants with increasing travel distances, showing that 50% of the greenhouse gas (GHG) emissions for all participants' trips result from long-distance travels with a round-trip distance of over 10,000 km. **c** Environmental profiles of the virtual conference and in-person conference with only one hub, suggesting the environmental and energy sustainability of virtual conference in all impact categories. Colors represent different processes within the life cycle stages. Specifically, transitioning from in-person to virtual conferencing reduces the carbon footprint by 94% and cumulative energy demand (CED) by 90%. Among the impact categories at the midpoint level, air transportation dominates fossil depletion, marine eutrophication, natural land transformation, ozone depletion, and photochemical oxidant formation for the 1-hub in-person scenario. And food preparation, electricity consumption at home, and information communication technology (ICT) services contribute to the majority of each impact category for the baseline virtual scenario. Moreover, agricultural land occupation, terrestrial ecotoxicity, and water depletion are dominated by food preparation for both in-person and virtual conferences. CO2 eq., CO2 equivalent.

butter production contribute to most agricultural land occupation, as cattle production occupies far more land than other agricultural products[32]. On the other hand, cattle production requires intensive water inputs[32], similar to the production of some grains considered in this study, such as almonds, rice, and wheat[32,33]. Moreover, terrestrial ecotoxicity is associated with pesticide residues and fertilizer losses for the agricultural systems[34]. The environmental impacts of food preparation may vary with dietary types, and further reductions can be achieved by switching to a more plant-based diet (Supplementary Fig. 2).

Besides food preparation, electricity consumption for accommodation during the virtual conference and the video-conferencing contribute considerably to most impact categories. Notably, the environmental profile of electricity production is highly dependent on the geographical distribution of participants. The result indicates that the environmental benefits of virtual conferences are weakened if a large number of participants are located in regions without a strong penetration of renewables in the power grid. Nevertheless, with the increasing share of renewable energy in the power grid worldwide, gradual reductions in the environmental impacts of electricity production are promising.

ICT can be identified as another environmental hotspot. This is because electricity consumption of ICT is around half of the electricity used at home, and electricity consumption accounts for the majority of ICT in most impact categories. Exceptionally, ICT dominates metal depletion because of the intensive metal use during computer production. The environmental performance of ICT can be improved by improving the energy efficiency of devices, networks, and data centers.

**Impacts of conference hubs on environmental sustainability.** This section compares life cycle carbon footprint, CED, and full-spectrum environmental profiles of the multi-hub in-person conferences and the baseline virtual conference. As shown in Fig. 2c, transitioning from in-person to virtual conferencing reduces the carbon footprint by 94% and CED by 90%, primarily due to the difference in transportation emissions of these two scenarios. To be more specific, among the GHG emissions of 280 t $CO_2$ equivalent for all participants' trips, 50% of which is resulted from long-distance travel with a round-trip distance of over 10,000-km, as shown in Fig. 2b. Owing to the landing and take-off cycle (LTO) emissions of flights, environmental impacts of air transportation scale with distance through a polynomial function, with greater distance tending to have lower unit environmental impacts (Supplementary Fig. 3 and Supplementary Table 3). Through calculation, for a single passenger, the carbon footprint of driving is equivalent to that of air transportation when the travel distance is around 500 km[35,36]. Therefore, the unit carbon footprint for trips primarily by plane tends to be smaller than that for driving, as shown in Fig. 2a. The trips primarily by airplanes embrace not only air transportation but also ground transportation from the participants' origins to their nearest airports and from the destination airports to the conference hubs. The carbon footprint of trips primarily by airplanes increases with longer driving distances, suggested by the outliers in Fig. 2a.

Besides the carbon footprint and CED, air transportation dominates several impact categories at the midpoint level, including fossil depletion, marine eutrophication, natural land transformation, ozone depletion, and photochemical oxidant formation (Fig. 2c). In order to model the entire life cycle of air transportation, the environmental impacts involve aircraft production, airport construction, fuel production, and exhaust emissions from aircraft operation[35]. In particular, kerosene production as the fuel of aircraft results in the dominance of air transportation in CED, ozone depletion, natural land transformation, and fossil depletion. An entire flight consists of LTO and climb-cruise-descent (CCD) phases. Air transportation dominates carbon footprint through exhaust emission of GHGs during the entire flight and aviation-induced cloudiness during the CCD phase. Direct NOx emission during the entire flight contributes to the majority of marine eutrophication. The leading contributors for photochemical oxidant formation are aircraft construction, fuel production, and LTO emissions of unburned hydrocarbons and NOx. Among other impact categories, freshwater ecotoxicity, marine ecotoxicity, metal depletion, and terrestrial acidification are dominated by car transportation. The dominance is primarily related to the vehicle infrastructure, vehicle maintenance, and fuel consumption[36], which are included in the environmental impacts of ground transportation by car. Moreover, agricultural land occupation, terrestrial ecotoxicity, and water depletion are dominated by food preparation for both in-person and virtual conferences. The reasons are discussed in the previous section.

Food preparation, electricity consumption at home, and ICT services contribute to the majority of each impact category for the baseline virtual scenario (Fig. 2c). However, the transitioning from in-person to virtual conference reduces the environmental impacts by only 35–51% in some impact categories, including agricultural land occupation, terrestrial ecotoxicity, and water depletion. The reason is that transportation is less predominant, and food preparation contributes the most to agricultural land occupation, terrestrial acidification, terrestrial ecotoxicity, and water depletion categories of environmental impacts for both in-person and virtual conferences. Furthermore, transitioning from in-person conference to pure virtual mode increases the environmental impacts of electricity consumption by 50% for agricultural land occupation and 30% for human toxicity. This result can be attributed to the geographic concentration of virtual participants in regions where biomass-based and coal-fired electricity accounts for a higher share of electricity production compared to the power grid energy sources in the 1-hub in-person conference scenario (Supplementary Fig. 4)[37,38]. The result indicates that the environmental benefits of virtual conferences are weakened if a large number of participants are located in regions without a strong penetration of renewables in the electric power grid.

The environmental benefits of in-person scenarios vary considerably as the number of conference hubs increases. Figure 3a–f presents the selection of conference hubs and the assignment of participants to their nearest conference hubs by travel distance. There is at least one U.S. hub for all in-person scenarios because most participants are from North America, especially the U.S. (77%). When there is more than one hub, a European hub is selected to accommodate 9% of the participants who come from Europe and the Middle East. The number of U.S. hubs increases for scenarios with more than two hubs. We also compare the environmental impacts across the in-person scenarios, as shown in Fig. 4. The result suggests that adding conference hubs can reduce the carbon footprint and CED of the in-person conference by half (Fig. 4a, b). Nevertheless, the carbon footprint and CED of a 6-hub in-person conference are still 739% and 637% higher than those of a virtual conference, respectively. Adding more hubs leads to reductions in the air transportation distances, as shown in Fig. 4c. Therefore, impact categories led by air transportation achieve the most reduction benefits from the addition of conference hubs. However, the changes in the ground transportation distance across these in-person scenarios are subtle. Specifically, as the total number of hubs grows, the driving distances increase since more participants switch from air

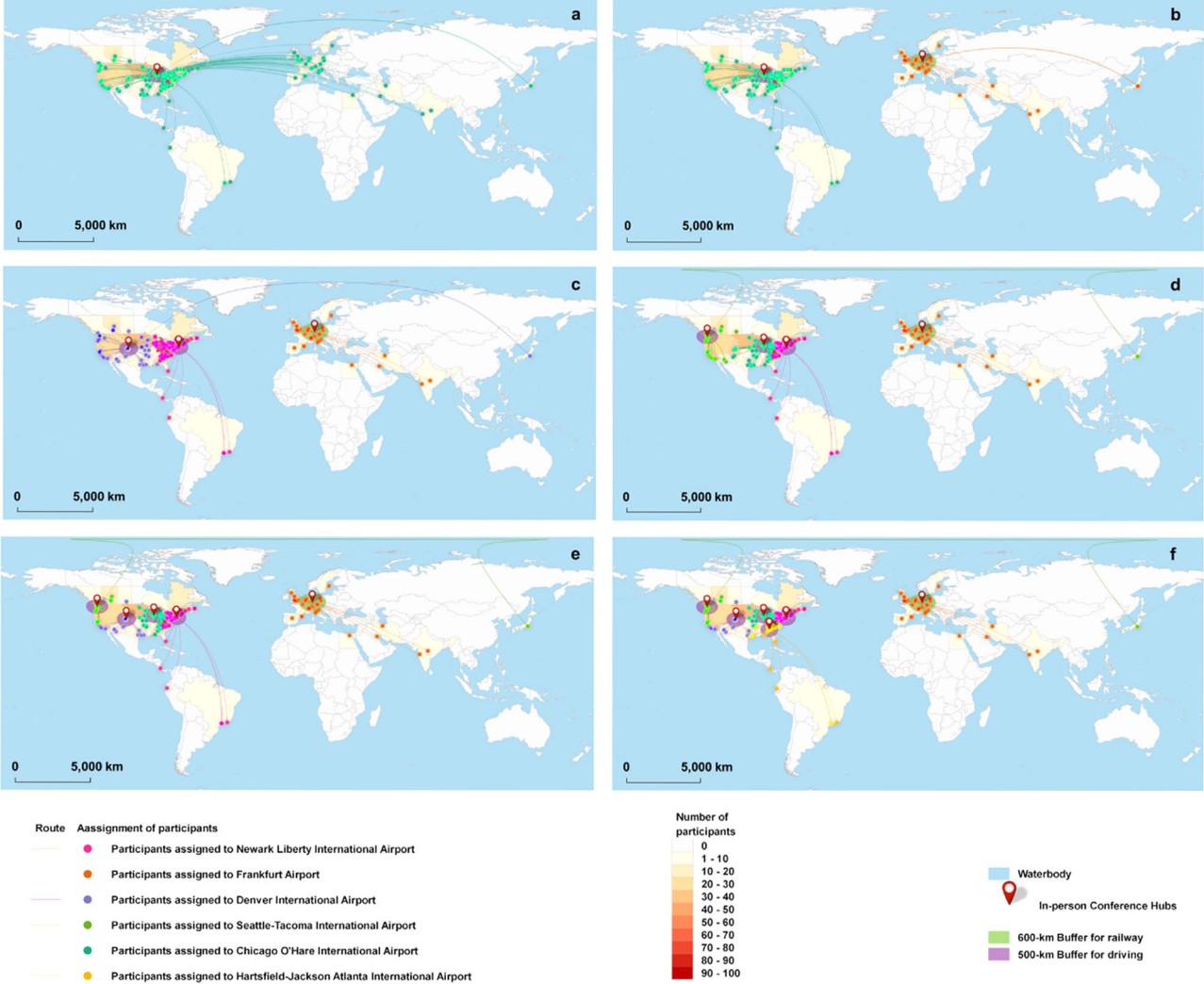

**Fig. 3 Map of transportation routes and hub decisions for in-person scenarios. a** 1-hub in-person conference. **b** 2-hub in-person conference. **c** 3-hub in-person conference. **d** 4-hub in-person conference. **e** 5-hub in-person conference. **f** 6-hub in-person conference. Dots represent the origins of participants, and lines represent the route from their origins to their assigned conference hubs. Their colors indicate the assignments of participants to the conference hubs. Color scale shows the number of participants in each region around the world. These regions are divided according to the regional datasets for electricity production in Ecoinvent[39].

transportation to driving. However, the addition of conference hubs also induces the reassignments of participants to a closer hub, which subsequently reduces the driving distances. When the number of hubs continues to grow, less modal switching and participants' reassignments occur, and therefore the driving distances become steady. Due to the greatest driving distances, the 6-hub in-person scenario has the highest environmental impact among all in-person scenarios in multiple impact categories, including freshwater ecotoxicity, marine ecotoxicity, and terrestrial acidification, which are all contributed significantly by car transportation. In addition, the environmental impacts of several impact categories, including freshwater eutrophication, human toxicity, ionizing radiation, particulate matter formation, and terrestrial acidification, are susceptible to the spatial variation of electricity production by energy sources and technologies. Hence, the choice of hub locations is critical to these midpoint indicators.

**Trade-offs between in-person communication and carbon footprint**. In addition to the environmental benefits of adding

conference hubs, we investigate the trade-offs between in-person interactions and the carbon footprint of hybrid conferences. Two sets of hybrid scenarios are considered based on their specific constraints:

- MTD scenarios: the maximum one-way travel distance (1 000 km, 3 000 km, 5 000 km, or 10,000 km), and the number of conference hubs.
- MVP scenarios: the maximum virtual participation level (10%, 30%, 50%, or 70%) and the number of conference hubs.

The number of conference hubs varies from one to six due to the need to minimize medium- to long-haul flights of over 1000 km[40]. Moreover, in-person participation is prioritized by the MTD scenarios. Since most participants are from Europe and North America, the maximum one-way travel distance of 1000 km, 3000 km, 5000 km, and 10,000 km is considered for the size of states in the U.S., European countries, continental Europe, and North America, respectively. Notably, similar to the in-person scenarios, the hybrid scenarios make rational decisions on the hub locations and the participant assignments based on

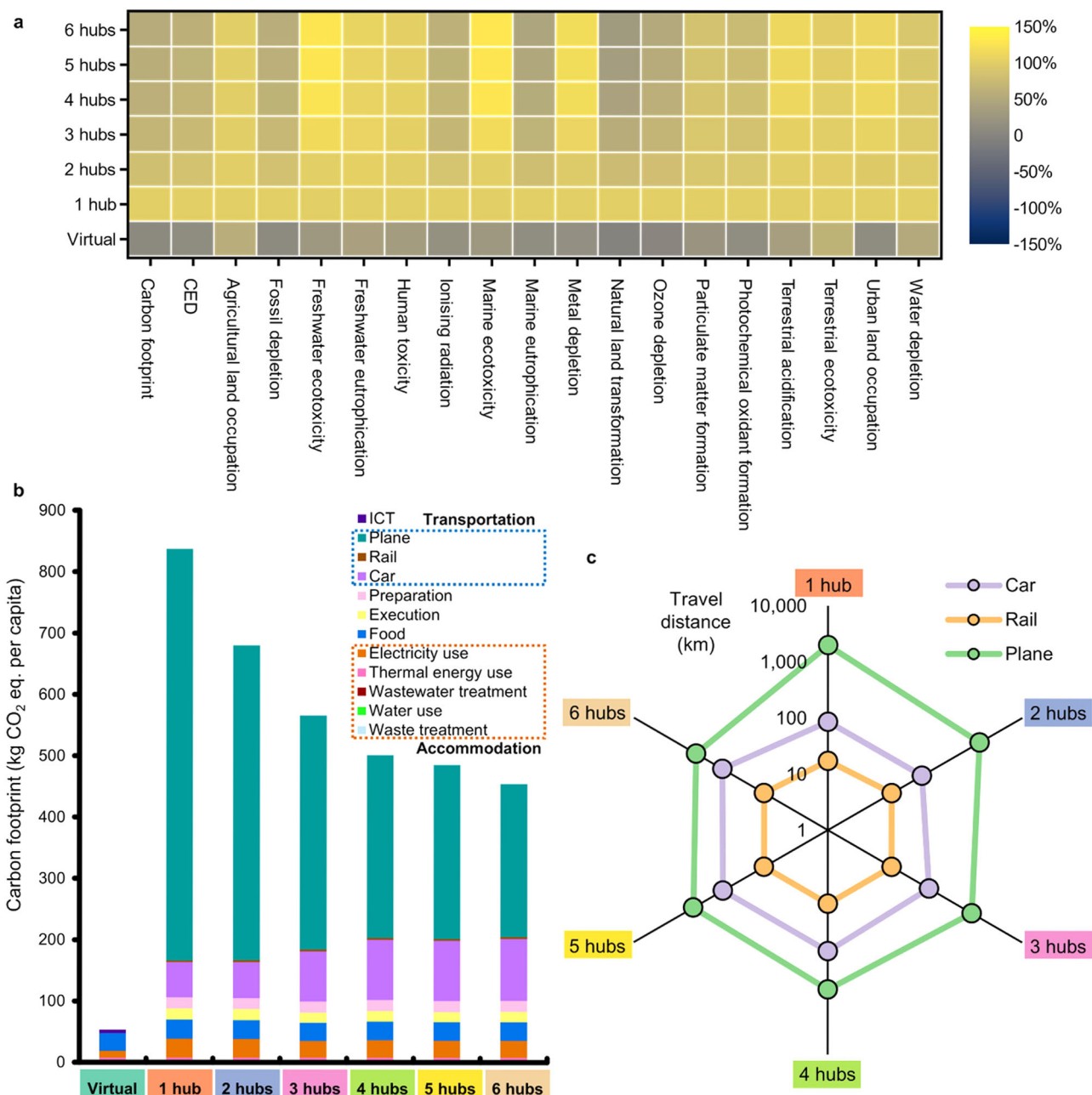

**Fig. 4 Overall environmental sustainability of the virtual and in-person conferences. a** Carbon footprint, Cumulative energy demand (CED), and 17 ReCiPe midpoint indicators of the in-person scenarios, including the virtual conference and in-person conferences with 1 to 6 hubs, for an average participant. Colors indicate the intensity levels of environmental impacts. **b** Breakdowns of carbon footprint per participant for the in-person scenarios. Colors represent different processes within the life cycle stages. **c** Average travel distances by transportation mode on a logarithmic scale for the in-person scenarios with 1 to 6 hubs. $CO_2$ eq., $CO_2$ equivalent.

the optimization results for minimizing the total travel distances instead of realistic data. Therefore, the results of these hypothetic scenarios can be considered as an upper bound for the carbon footprint of realistic hybrid conferences.

Similar to the in-person scenarios shown in Fig. 4, the addition of more hubs for hybrid conferences results in a reduction of transportation distances, and consequently, a lower carbon footprint. As shown in Fig. 5a, raising the virtual participation level to 50% can cut the carbon footprint of pure in-person conferences by two-thirds for all scenarios with the same number of hubs. For multi-hub MVP scenarios, raising the virtual participation level to 70% eliminates air travel, resulting in an

over 80% reduction in the carbon footprint. The MTD scenarios show greatly diminished virtual participation. As shown in Fig. 5b, the scenarios with maximum travel distances of 3 000 km, 5 000 km, and 10,000 km keep the virtual participation levels below 4% for the multi-hub scenarios and below 13% for the 1-hub scenarios. With fewer conference hubs and a larger maximum travel distance, the number of intercontinental flights and the average travel distance greatly increase to compromise the prioritized in-person participation (Supplementary Fig. 5–12). Consequently, these MTD scenarios have higher carbon footprints than that of the MVP scenarios and even in-person scenarios. In detail, the carbon footprint is up to 1 799 kg $CO_2$

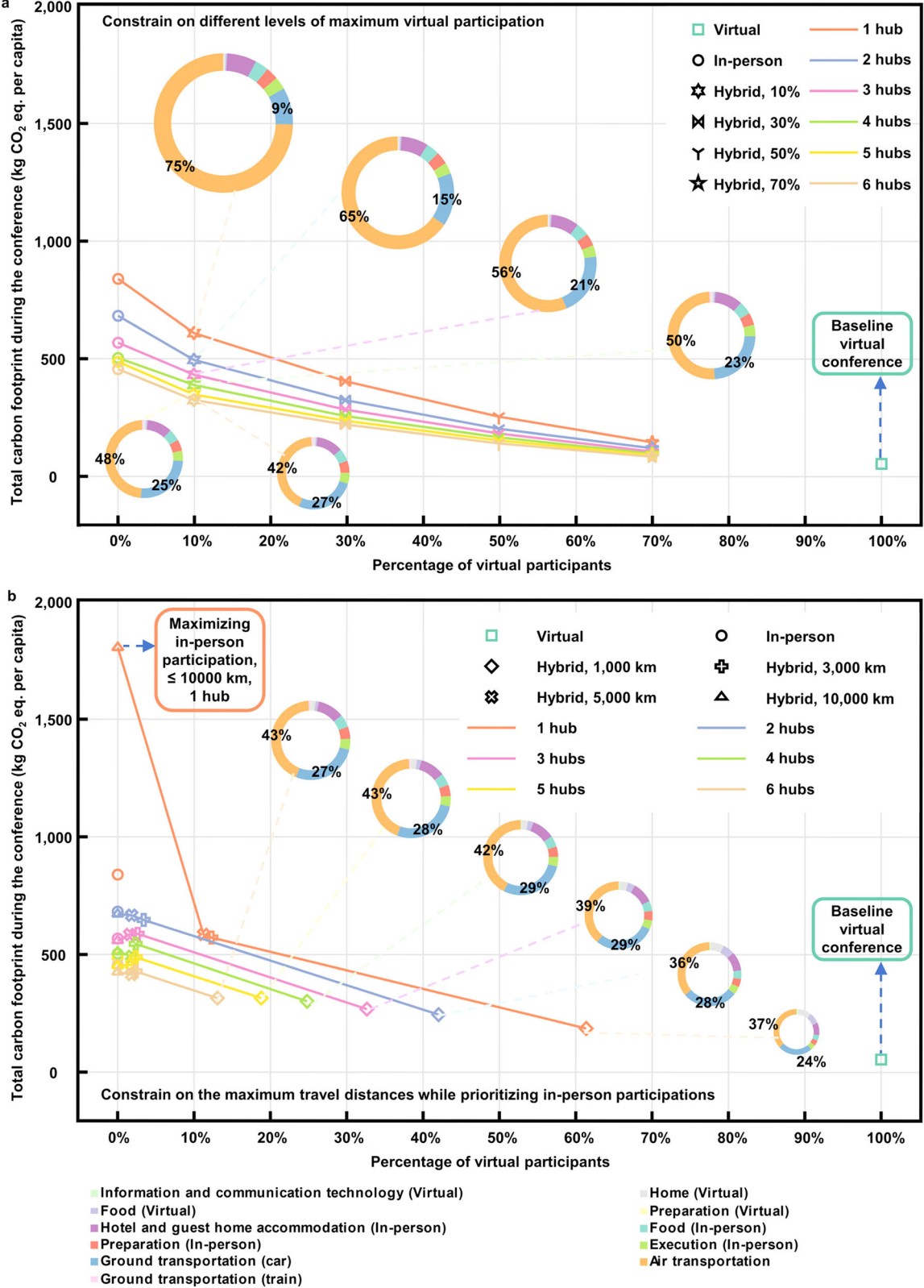

**Fig. 5 Trade-offs between face-to-face communication and carbon footprint.** Different markers indicate the results of different scenarios, and their colors refer to the number of conference hubs. Doughnut charts represent the breakdowns of scenarios, pointed by the blue dashed lines. **a** Comparison between the hybrid scenarios with a constraint of the maximum virtual participation, the in-person scenarios, and the baseline virtual scenario. **b** Comparison between the hybrid scenarios with a constraint of the maximum travel distance, the in-person scenarios, and the baseline virtual scenario. $CO_2$ eq., $CO_2$ equivalent.

equivalent per participant, approximately twice of that for the 1-hub in-person scenario. However, with a relatively small maximum travel distance (e.g., < 3000 km in this study) or more conference hubs, MTD scenarios achieve substantial reductions in the carbon footprint. The result suggests that setting a relatively small maximum travel distance is the most effective in reducing the carbon footprint of hybrid conferences. In this study, the maximum travel distance should be set below 3000 km. It is worth mentioning that this threshold is subject to the number of hubs and changes in the geospatial distribution of participants. For instance, if participants are mainly concentrated in the West Coast of the U.S. and minorly distributed in the East Coast of the U.S., setting the maximum travel distance as 3000 km may have a similar effect as setting the maximum travel distance as 10,000 km for this study. The results for other impact categories, including CED and 17 ReCiPe midpoint indicators, generally follow similar trends as the carbon footprint results. Details are shown in Supplementary Fig. 13–14.

## Discussion

We perform a holistic LCA to compare the environmental sustainability across virtual, in-person, and hybrid conferences, which also takes into consideration the number of hubs and in-person participation levels. As summarized in Supplementary Fig. 1, the carbon footprint of the 1-hub in-person conference reported in this work (840 kg $CO_2$ equivalent per participant) is consistent with the previous LCA results on in-person conferences[4–13] and is comparable to the monthly carbon footprint of an average U.S. citizen in 2018[41]. Our results show that switching from in-person conference to pure virtual mode substantially reduces the carbon footprint by 94% and CED by 90%. As we consider the virtual conference's whole life cycle, less reduction in carbon footprint can be achieved by switching to full virtual or hybrid mode, compared to values reported by other studies[4,5,13,26]. Adding more conference hubs lowers the average air transportation distance and increases the average travel distance by car at a decreasing rate. Due to the increasing travel distances by car, the environmental impact of the 6-hub in-person scenario is higher than that of the other in-person scenarios in several impact categories, especially those contributed significantly by car transportation, including freshwater ecotoxicity, marine ecotoxicity, and terrestrial acidification. Our analysis provides insight into the potentials for future conferences to balance between mitigating their carbon footprints and maintaining a considerable level of in-person interactions. We examine the trade-offs between in-person interactions and the carbon footprint of hybrid conferences with constraints either on the maximum travel distance or the maximum virtual participation. The carbon footprint mitigation potential of hybrid conferences is sensitive to the maximum travel distance. With a relatively large maximum travel distance, the hybrid conference cannot effectively mitigate the carbon footprint of its in-person counterpart. On the contrary, by imposing a constraint on the maximum virtual participation level, the Pareto-optimal solutions clearly illustrate trade-offs between the carbon footprint and level of in-person communications. Our results demonstrate that spatially optimal hubs for the hybrid conferences have the potential to slash carbon footprint and energy use by 60–70% while maintaining <50% of virtual participation. Therefore, from the environmental perspective, it is beneficial to hold a hybrid conference and decide the location of the hubs using the registration information or survey responses. The hub locations can be sub-optimal because differences may exist between the pre-conference information and the actual attendance. Adding hubs and increasing virtual participation levels tend to provide more environmental benefits, but this benefit becomes less prominent as the number of hubs and virtual participation level is high

enough. It is therefore important for conference organizers to consider the trade-off between organizational challenges and environmental sustainability.

In addition to the holistic LCA analysis, sensitivity analyses are performed to evaluate the uncertainty of the carbon footprint results in response to the variations in parameters of in-person and virtual conferences (Supplementary Fig. 2, 15–19). The carbon footprint of in-person scenarios is highly susceptible to the selection of characterization factors for air transportation and air transportation distances (Supplementary Fig. 19 and Supplementary Table 4). With more conference hubs, the carbon footprint becomes less variable to changes in these parameters. This result suggests that participants should make efforts to take flights with as few stopovers as possible. In terms of virtual conferences, intuitively, the most sensitive parameter for a virtual conference is daily participation, as the decline of daily participation takes away all environmental impacts related to those missing participants. However, diminishing synchronous virtual participation may add difficulty to communication and collaboration for virtual conferences that do not provide sufficient asynchronous attendance options, such as the recording proceedings[42]. In order to create more opportunities for exchange and follow-up discussion, virtual conference organizers are encouraged to provide contact information, recording proceedings, and electronic documents submitted by the presenters to all participants. These asynchronous attendance practices could support virtual conferences to improve equity, diversity, inclusivity, networking, early-career research promotion, and career development[18,21–23]. On the other hand, post-conference activities, such as downloading and playing asynchronous recordings, sending follow-up emails, and searching for materials, can result in environmental impacts. The relationship between daily virtual participation and the amount of post-conference activities is unknown. Therefore, we cannot expand our system boundary to account for the variation in the environmental impacts of these consequential activities. Future work could further explore the consequential environmental impacts of the rapidly expanding video-conferencing industry. Dietary type as well as the electricity and food consumption rate are the next most sensitive variable for a virtual conference. The ovo vegetarian diet performs better in mitigating carbon footprint than other types of diets, especially the stricter vegan diet. This is because the ovo vegetarian diet allows for egg consumption, of which the carbon footprint is lower than that of soybean products, while the vegan diet does not include eggs. For virtual conference organizers, advocating energy saving from heating and other residential electricity use (e.g., air conditioning, lighting, electronics, and appliances), food waste reduction, ovo vegetarian diet can be effective practices to improve sustainability. Internet and computer network-related electricity consumption is also a sensitive parameter as it accounts for around two-thirds of the carbon footprint associated with ICT (Supplementary Fig. 16). By 2030, the improvement in the energy efficiency of the network could result in a reduction of 2.9–5.9% in carbon footprint per participant, depending on the annual energy efficiency improvement rate of the worst- or best-case scenario[43].

## Methods

**Methodology overview**. We integrate the LCA and spatial analysis to investigate the environmental sustainability of conferences. The 2020 American Center for Life Cycle Assessment virtual conference is presented as a case study for all scenarios. Based on the virtual conference participants' data, scenario analysis and facility location optimization are combined to understand the environmental benefits of multiple conference hubs and the trade-offs between face-to-face communication and carbon footprint. Furthermore, sensitivity analyses on the parameters of ICT and dietary type are conducted to evaluate the impacts of parameter uncertainty and seek further carbon footprint mitigation. There are 536 virtual participants in total, and geographic information of 383 participants

is available. The per capita environmental impacts are then estimated based on the 383 participants.

The LCA approach is conducted in four phases, including goal and scope definition, LCI, LCIA, and interpretation, described in the following.

**Goal and scope definition**. In this work, we aim to evaluate the life cycle environmental impacts of conferences with different modes. The scope of this LCA focuses on the stages of food preparation, accommodation, preparation, execution, information and communication technology, and transportation. Post-conference activities of participants, such as downloading and playing asynchronous recordings, sending follow-up emails, and searching for materials, are out of the scope of this attributional LCA. We define one average participant as the functional unit following existing LCA studies on conferences for a fair comparison across different scenarios[5,7]. Carbon footprint, CED, and 17 ReCiPe midpoint indicators from the hierarchist perspective are adopted to examine the burdens in different environmental impact categories, including agricultural land occupation, fossil depletion, freshwater ecotoxicity, freshwater eutrophication, human toxicity, ionizing radiation, marine ecotoxicity, marine eutrophication, metal depletion, natural land transformation, ozone depletion, particulate matter formation, photochemical oxidant formation, terrestrial acidification, terrestrial ecotoxicity, urban land occupation, and water depletion[27]. The following subsections describe the LCI phase of LCA that quantifies and compiles all the mass and energy flows entering and exiting the system boundary. The recycled content approach is used for the LCI modeling[44].

**Life cycle inventory for food and accommodation**. We assess the environmental impacts of food and accommodation during the conference. The per-participant food supply by country from FAOSTAT is used as representatives for the amount and composition of meals consumed during the conference[30,31]. Major food categories include fruits, vegetables, grain, dairy, protein food, legumes, vegetable oil, animal fats, and sugars. Due to the lack of LCIA data, multiple representatives for each food category are chosen, and the amount of the representative food is listed in Supplementary Table 5. According to the Ecoinvent database[39], food losses during distribution and preparation are included in the system boundary. Moreover, utilities for food preparation (i.e., electricity, thermal energy, and water) are considered and presented in Table 1[6]. It is worth mentioning that the treatment of food waste is not accounted for at the food preparation stage; instead, it is considered to be part of the waste treatment at the accommodation stage to avoid double counting.

The accommodation stage only considers utilities and waste treatment for staying at a hotel or guest home. Specifically, electricity consumption for a guest home is estimated from the U.S. Energy Information Administration's State Energy Data System[45]. The electricity consumption for hotels, thermal energy and water consumption for both guest homes and hotels are estimated from previous studies (Table 2)[46,47]. Waste composition and management pathways are obtained from the U.S. Environmental Protection Agency and summarized in Table 3[48]. Other fast-moving consumer goods, such as soaps, shampoo, conditioner, and tissue, as well as the losses in slow-moving consumer goods, such as furniture and appliances, are considered beyond the scope of this study, following existing LCA of hotels[47,49]. The recycled content approach is adopted for the end-of-life modeling, by which secondary material bears no environmental impacts from the

activities related to the life cycle of primary material, and no credits are given to the material recycling[44].

**Life cycle inventory for preparation and execution**. The preparation stage involves preparation activities before the conference associated with permanent conference committees, local conference committees, secretariat, website, printed materials, and souvenirs, following the setting considered in the literature[6]. In particular, activities of permanent conference committees, local conference committees, secretariat, and participants consist of printing and computer usage. Notably, all activities related to printing are involved, including production of paper and toner module, printer operation (i.e., electricity consumption), and disposal of waste paper and used toner module. Similarly, computer usage includes the production and operations of computers, asymmetric digital subscriber line (ADSL) modems, and Internet protocol (IP) routers. In addition, computer usage is considered for website maintenance. Booklets (conference program and proceedings) and jute bags (souvenirs) are also prepared before the conference. Other activities of committee members and secretariat indirectly related to the conference preparation, such as transportation, food, and accommodation, are neglected because they are outside the scope of this study. For the execution stage, utilities (electricity and water consumption) and waste treatment of the conference venue are included in the system boundary[50]. The waste composition is set as the composition of municipal solid waste considered in the accommodation stage. Materials and energy inventories of the preparation and execution stage are summarized in Table 4.

**Life cycle inventory for information and communication technology**. To estimate the environmental impacts of virtual conferences, we incorporate an environmental impact assessment framework of Internet services[51]. The system boundary of the ICT stage consists of infrastructure, network, and server related to video-conferencing. Specifically, four analysis layers are examined, including the energy consumption of the infrastructure, the shares of access network traffic and shares of IP protocols delivering the investigated Internet services, shares of traffic classes representing the end-user activities, and shares of the investigated Internet services in each traffic class. As participants of conferences usually use search engines or other office software simultaneously on the computer, mobile devices are excluded from the system boundary. Energy consumption of infrastructure, including router, Internet access equipment, computer, and data center equipment (cooling systems, lighting, and power supplies), are considered following previous studies[39,52].

In addition, the production and distribution of routers, Internet access equipment, and computer are included in the system boundary. At the same time, the construction of data center infrastructure is excluded because of the unavailable LCIA data and dominance of the operational phase to the overall environmental impacts[53]. The energy intensity of the network for the video traffic class ($EI_{network}$) is calculated as follows:

$$EI_{network} = \frac{\lambda_{fixedline} \cdot \lambda_{IPv4} \cdot \lambda_{TCP} \cdot \lambda_{HTTP} \cdot \lambda_{video} \cdot \sum_{k \in K} E_k}{DT_{video}} \quad (1)$$

where $\lambda_{fixed\ line}$, $\lambda_{IPv4}$, $\lambda_{TCP}$, $\lambda_{HTTP}$, $\lambda_{video}$ represent the share of fixed line traffic in the total IP traffic, the share of IPv4 based traffic in the total IP traffic, the share of transmission control protocol (TCP) based traffic in the total IP traffic, the share of hypertext transfer protocol (HTTP) based traffic in the TCP based traffic, the share of video traffic class in the HTTP based traffic, respectively. $K$ indicates the set of network components, including packet switched core, fixed line customer premises equipment, operator data center, office networks, and Internet core. $E_k$ denotes the energy consumption of each component of the network. $DT_{video}$ represents the fixed line traffic for the video traffic class. As the best available data of the network's energy consumption and data traffic is from 2016, an annual electricity efficiency improvement of 10% is considered following the setting of the expected scenario in the previous work[43]. The energy intensity of the server ($EI_{server}$) is estimated from the best available data of a 2015 Sweden study on the ICT sector[54] and extrapolated to the value for 2020 with an annual electricity efficiency improvement of 10%[43]. The calculation is as follows:

$$EI_{server} = \frac{E_{server}}{DT_{server}} \quad (2)$$

where $E_{server}$ represents the total energy consumption of servers, and $DT_{server}$ represents the total data traffic of the server. Data traffic of the virtual conference is computed following the survey of a recent study, which recognized 80% of participants attending the conference each day with a daily online duration of 5.5 h[5]. The energy consumption related to the network and server for the virtual conference ($VE_{network}$ and $VE_{server}$) is computed following Eqs. (3) and (4).

$$VE_{network} = EI_{network} \cdot DT_{virtual} \quad (3)$$

$$VE_{server} = EI_{server} \cdot DT_{virtual} \quad (4)$$

where $DT_{virtual}$ is the data traffic of the virtual conference calculated by multiplying the bandwidth of the video-conferencing with the total amount of online time for all participants. Downstream and upstream bandwidth of a Zoom group video calling for 720 high-definition videos is 1.8 and 2.6 megabits per second (Mbps), as

## Table 1 Energy and water consumption of food preparation per capita per day[6].

| Item | Value | Unit |
|---|---|---|
| Electricity | 4.00 | kWh |
| Heat | 6.00 | MJ |
| Water | 28.00 | kg |
| Wastewater | 0.028 | m³ |

## Table 2 Energy and water consumption of accommodation per capita per day by the type of accommodation[45–47,64,65].

| Type of accommodation | Item | Value | Unit |
|---|---|---|---|
| Guest home | Electricity | 12.02 | kWh |
| | Heat | 49.25 | MJ |
| | Water | 217.3 | kg |
| | Wastewater | 0.22 | m³ |
| Hotel | Electricity | 18.10 | kWh |
| | Heat | 18.36 | MJ |
| | Water | 300.00 | kg |
| | Wastewater | 0.30 | m³ |

**Table 3 Average amount, composition, and management pathways of municipal solid waste (MSW) in the U.S. per capita per day[48]. Blanks represent missing values.**

| Category | 2018 MSW generation (kg) | Weight recycled (kg) | Weight composted (kg) | Weight other food management pathways (kg) | Weight combusted with energy recovery (kg) | Weight landfilled (kg) |
|---|---|---|---|---|---|---|
| Paper and paperboard | 0.56 | 0.38 | | | 0.04 | 0.14 |
| Glass | 0.10 | 0.03 | | | 0.01 | 0.06 |
| Steel | 0.16 | 0.05 | | | 0.02 | 0.09 |
| Aluminium, other nonferrous metals | 0.05 | 0.02 | | | 0.01 | 0.03 |
| Plastics | 0.30 | 0.03 | | | 0.05 | 0.23 |
| Rubber and leather | 0.08 | 0.01 | | | 0.02 | 0.04 |
| Textiles | 0.14 | 0.02 | | | 0.03 | 0.09 |
| Wood | 0.15 | 0.03 | | | 0.02 | 0.10 |
| Food, yard trimmings, other MSW organics | 0.83 | | 0.21 | 0.15 | 0.08 | 0.38 |
| Other MSW | 0.07 | 0.01 | | | 0.01 | 0.05 |
| Total | 2.45 | 0.58 | 0.21 | 0.15 | 0.29 | 1.22 |

**Table 4 Material and energy consumption of preparation and execution[6,13,50].**

| Category | Item | Specification | Value | Unit |
|---|---|---|---|---|
| Preparation - local conference committee | Computer usage | Operation, computer, laptop, 68% active work | 2.00E−01 | h |
| | | Operation, computer, laptop, 23% active work | 2.00E−01 | h |
| | | Operation, computer, laptop, standby/ sleep mode | 2.00E−01 | h |
| | Printed matter, abstract reviewing process | Graphic paper, 100% recycled | 7.04E−03 | kg |
| | | Operation of printer, laser, black and white, per kg printed matter | 7.04E−03 | kg |
| | Printed matter, meeting | Graphic paper, 100% recycled | 3.00E−03 | kg |
| | | Operation of printer, laser, black and white, per kg printed matter | 3.00E−03 | kg |
| | Waste paper | Paper waste disposal | 1.00E−02 | kg |
| Preparation - permanent conference committee | Computer usage | Operation, computer, laptop, 68% active work | 1.40E−01 | h |
| | | Operation, computer, laptop, 23% active work | 1.40E−01 | h |
| | | Operation, computer, laptop, standby/ sleep mode | 1.40E−01 | h |
| | Operation of printer, laser, black and white, per kg printed matter | Graphic paper, 100% recycled | 1.31E−03 | kg |
| | | Operation of printer, laser, black and white, per kg printed matter | 1.31E−03 | kg |
| | Waste paper | Paper waste disposal | 1.31E−03 | kg |
| Preparation - participants | Abstract writing | Operation, computer, laptop, active mode | 7.05E−01 | h |
| | Poster printing | Paper | 4.65E−03 | kg |
| | | Operation, printer, laser, color, per kg printed paper | 4.65E−03 | kg |
| | Slide preparation | Operation, computer, laptop, active mode | 5.35E−01 | h |
| Preparation - others | Office work - secretariat | Operation, computer, laptop, active mode | 2.70E+00 | h |
| | Website maintenance | Operation, computer, laptop, active mode | 1.62E+00 | h |
| | Conference program | Paper, wood containing, supercalendered | 3.86E−01 | kg |
| | | Operation, printer, laser, color, per kg printed paper | 3.86E−01 | kg |
| | Proceedings booklet | Paper, wood containing, supercalendered | 3.41E+00 | kg |
| | | Operation, printer, laser, color, per kg printed paper | 3.41E+00 | kg |
| | Jute bag | Jute textile production | 7.97E−02 | kg |
| | | Transport, cargo aircraft, intercontinental | 5.90E−01 | ton-km |
| | | Transport, freight, truck | 2.39E−02 | ton-km |
| Execution - Conference meeting rooms | Electricity | Electricity production | 2.84E+01 | kWh |
| | Water | Tap water production | 8.68E+01 | kg |
| | Wastewater | Wastewater treatment | 8.68E−02 | m$^3$ |
| | Waste | Waste treatment | 1.22E+00 | kg |

**Table 5 Material and energy consumption of the information and communication technology for virtual-conferencing[5,39,43,51,52,54,55].**

| Category | Item | Value | Unit |
|---|---|---|---|
| Device | Router, internet | 1.09E−01 | unit |
| | Internet access equipment | 1.03E+00 | unit |
| | Computer, laptop | 1.54E+00 | unit |
| Device-related electricity | Energy consumption for router | 0.00E+00 | kWh |
| | Energy consumption for internet access equipment (ADSL, DSLAM) | 0.00E+00 | kWh |
| | Energy consumption for laptop, video mode | 3.43E+02 | kWh |
| Network-related electricity | PS-Core | 2.38E+00 | kWh |
| | Fixed line CPE | 3.83E+00 | kWh |
| | Operator DC | 4.75E−01 | kWh |
| | Office networks | 1.18E+00 | kWh |
| | Internet core | 2.88E−01 | kWh |
| Server-related electricity | Infrastructure | 6.67E−01 | kWh |
| | Network | 5.56E−02 | kWh |
| | Storage | 2.22E−01 | kWh |
| | Server | 1.07E+00 | kWh |

obtained from Zoom[55]. Following the 2020 projection with expected improvement in energy efficiency from a previous study[52], the energy consumption of data center can be broken down into four equipment categories, namely infrastructure (33%), network (3%), storage (11%), and servers (53%). The material and energy inventories of the ICT stage are summarized in Table 5.

**Life cycle inventory for transportation.** To assess the environmental impacts of transportation, we calculate the distances between participants and the conference hubs. First, the participants' addresses are converted into geographic coordinates (longitude and latitude) using Google Geocoding API[56]. The travel distances for ground transportation are then computed using Google Distance Matrix API[57]. Moreover, great-circle distances are calculated for non-stop air transportation. Travel distances by rail for European participants and by car for participants outside Europe are calculated due to the passenger rail accessibility in Europe and passenger cars' popularity in the U.S.[58]. Notably, the travel distances by rail are computed based on the public transit routes[59]. Previous studies made decisions on the transportation modes of participants based on distance-based or time-based thresholds[6,12,26,29]. Since the travel time computed by Google Distance Matrix API is susceptible to the traffic, departure time, and route, we adopt the distance-based thresholds in this study. If the calculated travel distance for a one-way trip of a participant is >600 km by rail in Europe or 500 km by car in areas outside Europe, air transportation is selected as the primary transportation mode. The 600-km threshold for rail transport is considered following a previous work[12]. According to the U.S. Department of Transportation's modal choice over distance, we consider a 500-km threshold for driving[58]. Notably, the total transportation distance of a participant selecting air transportation as the primary transportation mode sums up the great-circle distance of air transportation, the travel distance between the participant's origin and its nearest airport, and the travel distance between the assigned conference hub and its nearest airport. The nearest airport is selected from the pool of the world's top 60 and U.S. top 60 busiest airports by passenger traffic[60]. For those participants whose origins are far away from all the airports in the pool, we select their nearest airports based on the available flight information on Google Flights.

**Life cycle impact assessment.** In this study, life cycle GHG emissions, direct and indirect energy use, and environmental impacts of a broad set of impact categories are demonstrated using carbon footprint, CED, and ReCiPe midpoint indicators, respectively. In this phase, characterization factors transform the long list of LCIs into environmental impacts for the investigated impact categories. Most of the characterization factors are collected from the Ecoinvent database[39]. The characterization factors for air transportation in Ecoinvent are classified based on four distance categories (i.e., very short-haul, short-haul, medium-haul, and long-haul flights). To be discriminative on the air transportation distances, we adopted the characterization factors from a comprehensive LCA study on air transportation[35]. Flights with 100–1 200 km transportation distances are reported, and characterization factors of air transportation from 1990 to 2050 are assessed and projected by Cox et al.[35]. We fit the characterization factors of different distances to polynomial functions for the year 2020 and summarized the parameters in Supplementary Table 2. Instead of being archived in the Ecoinvent database, characterization factors for some processes, such as egg and juice production, are obtained from the existing literature to fill the data gaps[61,62].

**Spatial analysis and optimization model.** To investigate the impacts of conference hubs on the environmental sustainability of the conference, a combination of spatial analysis and facility location optimization is employed. Conferences are usually held in locations that can meet the needs of participants, such as accessibility of transportation options. To best accommodate these needs, we selected the world's top 30 busiest airports by passenger traffic as alternative locations for the conference hubs[60]. Since 77% of the participants are from the U.S., more than one alternative location in the U.S. is expected to be chosen for scenarios allowing multiple conference hubs[60]. Therefore, we selected the world's top 30 and U.S. top 15 busiest airports by passenger traffic as alternative locations for the conference hubs for these multi-hub scenarios. A distance matrix of transportation distances between participants and the alternative locations is computed using the travel distances for ground transportation and great-circle distances for air transportation, as described in the Transportation section. The locations of conference hubs for each scenario are determined using the facility location optimization model as follows. The objective is to minimize the total transportation distances of all participants, shown in Eq. (5), where parameter $D_{i,j}$ denotes the transportation distance between participant $i$ and potential location $j$ for conference hubs, and binary variable $x_{i,j}$ serves as an indicator for the assignment of participant $i$ to the conference hub $j$. Each participant must be assigned to one of the alternative locations, as shown by constraint (6), where $I$ denotes the set of participants, indexed by $i$. Constraints (7), (8) determine whether the alternative location $j$ is selected as one of the conference hubs. $J$ represents the set of alternative locations for conference hubs, indexed by $j$; binary variable $y_j$ indicates whether the alternative location $j$ is selected as a conference hub; $M$ is a sufficiently large positive number applied for the big-M method; Parameter $\beta$ denotes the minimum number of participants assigned to a conference hub. $\beta$ is set as 20 for this study. Constraint (9) represents the number of conference hubs, of which the value varies by scenarios; the parameter $N$ represents the number of conference hubs. Constraints (10) and (11) define $x_{i,j}$ as a binary variable and $y_j$ as a non-negative continuous variable.

$$\min \sum_i \sum_j D_{i,j} x_{i,j} \tag{5}$$

$$s.t. \sum_j x_{i,j} = 1, \forall i \in I \tag{6}$$

$$\sum_i x_{i,j} \leq M \cdot y_j, \forall j \in J \tag{7}$$

$$\sum_i x_{i,j} \geq \beta \cdot y_j, \forall j \in J \tag{8}$$

$$\sum_j y_j = N \tag{9}$$

$$x_{i,j} \in \{0,1\}, \forall i \in I, j \in J \tag{10}$$

$$y_j \in \{0,1\}, \forall j \in J \tag{11}$$

For hybrid conference scenarios, either the maximum transportation distance is limited while maximizing participation, or the minimum level of in-person participation is specified. In the former case, the maximum one-way travel distance is set as 1000 km, 3000 km, 5000 km, or 10,000 km considering the size of continental Europe and North America, where most participants are from. The distance matrix components that are greater than the specified maximum transportation distance are replaced with a sufficiently large positive number. For the latter case, constraint (6) is replaced by an inequality as shown by constraint (12), as not all participants are necessarily assigned to a conference hub for a hybrid conference. In addition, constraint (13) is added to specify the minimum number of in-person participants, where parameter $\alpha$ denotes the percentage of in-person participation (i.e., 10%, 30%, 50%, or 70%) and parameter $P$ represents the total number of participants. $\alpha$ is set as 20 for this study.

$$\min \sum_i \sum_j D_{i,j} x_{i,j} \tag{5}$$

$$s.t. \text{ Constraints: } (7)-(11)$$

$$\sum_j x_{i,j} \leq 1, \forall i \in I \tag{12}$$

$$\sum_i \sum_j x_{i,j} \geq \alpha \cdot P \tag{13}$$

**Reporting summary.** Further information on research design is available in the Nature Research Reporting Summary linked to this article.

## Data availability

All data generated in this study have been provided in the Supplementary Information and deposited in GitHub [https://github.com/PEESEgroup/Virtual_Con] with the published version archived (https://doi.org/10.5281/zenodo.5515049)[63]. Microsoft Excel (version 2109), QGIS 3.16.3, and Python 3.7.3 were used to analyze data. In addition, the

LCIA data used in this study are available in the Ecoinvent database [https://ecoinvent.org/the-ecoinvent-database/], and the food supply data used in this study are available in the FAOSTAT database [https://www.fao.org/faostat/en/#data].

## Code availability

All data generated in this study have been provided in the Supplementary Information and deposited in GitHub [https://github.com/PEESEgroup/Virtual_Con] with the published version archived (https://doi.org/10.5281/zenodo.5515049)[63].

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

## Acknowledgements

This work is based upon work supported in part by the National Science Foundation (NSF) under Grant No. CBET-1643244 (to F.Y.). This work was supported in part by the Cornell Atkinson Center for Sustainability. J.J.K. acknowledges support from the EU project Sustainable Process Integration Laboratory – SPIL, funded as project No. CZ.02.1.01/0.0/0.0/15_003/0000456, by Czech Republic Operational Programme Research and Development, Education, Priority 1: Strengthening capacity for quality research operated by the Czech Ministry of Education, Youth and Sport.

## Author contributions

F.Y. and Y.T. contributed to the study design, data processing, analysis, and result interpretation. D.S and Y.T. collected the data. Y.T. and F.Y. wrote the manuscript. J.J.K. revised the manuscript. F.Y., D.S., J.J.K, and Y.T. reviewed the final manuscript.

## Competing interests

The authors declare no competing interests.
