## [Peer Review File · Nature Communications]

Reviewer comments, first round review -

Reviewer #1 (Remarks to the Author):

In this manuscript Tao et al performed a comparative life-cycle assessment of in-person, fully virtual and hybrid conferences and found that a switch to fully virtual conferences results in 94% reduction in carbon footprint. Additionally, the authors offer recommendations on dietary food items and energy efficiencies for further reductions in carbon footprint of conferences in various formats. This is an important, well-conceived and well-written piece with interesting analysis and findings that will be a valuable addition to the growing literature examining academic and non-academic conferences in all disciplines for scientists, conference organizers, policy makers and the public.

Comments: I would like to ask the authors to also address the following points:

1. Lines 42-44, the authors write: “Whether future conferences should return in-person, keep purely virtual, or evolve to a hybrid of both has been widely debated (19).” — the phrase “kept purely virtual” could improve to “kept fully virtual” to better describe the conference organization mode.

2. Lines 53-55, the authors write that: “By contrast, supporters of virtual conferences consider virtual interactions as a far more accessible, inclusive and sustainable counterpart to the traditional in-person format.” — a number of references that the authors may wish to use to support this statement (as the virtual conferences of 2020-2021 have improved equity and inclusivity in terms of gender, career stage, geography, disability status of attendees and speakers) are: <https://www.nature.com/articles/s41562-021-01067-y>, [https://www.cell.com/trends/cognitive-sciences/fulltext/S1364-6613\(21\)00009-7?rss=yes](https://www.cell.com/trends/cognitive-sciences/fulltext/S1364-6613(21)00009-7?rss=yes), <https://elifesciences.org/articles/62668> , <https://elifesciences.org/articles/57892>).

3. Lines 55-56, the authors write that: “As a compromise of the two, hybrid conferences may be a viable solution.” — A successful hybrid conference that can be referenced here to support this statement was held in 2019: (<https://www.nature.com/articles/d41586-019-03899-1> , a permanent website for that event: <https://careconferences.org/>

4. Lines 136-141, the authors state that: “Other than the food preparation, electricity consumption at home during the virtual conference contributes considerably to most impact categories. Notably, the environmental profile of electricity production is highly dependent on the geographical distribution of participants. The result indicates that the environmental benefits of virtual conferences are weakened if a large number of participants are located in regions without a strong penetration of renewables in the power grid.” — would the electricity consumption of attendees at virtual conferences be different from daily electricity consumption of academics in any location at work (daily electricity consumption of researchers working at their desktop or laptop computer remotely from home) or on campus/in office? Would the electricity consumption be different if most or some conference talks are watched in recorded format later (as opposed to live streaming)?

5. The size of conferences (number of attendees) is an important consideration for conference organizers assessing features of meetings and optimizing the format and organization. This is highly variable (hundreds to tens of thousands) and an important factor for consideration during organization of the legacy(in-person) conferences. Have the authors performed sensitivity analysis

on the impact of the number of attendees as a parameter under various scenarios with regards to conference format (in-person, hybrid, fully virtual) and their recommendations? It would be beneficial to include this to the manuscript text.

6. Lines 229-231, the authors state that: “Adding more hubs leads to reductions in the air transportation distances, as shown in Fig. 4c. Therefore, impact categories led by air transportation achieve the most reduction benefits from the addition of conference hubs.” and in lines 318-322: “Adding more conference hubs lowers the average air transportation distance and increases the average travel distance by car at a decreasing rate. Due to the increasing travel distances by car, the environmental impact of the 6-hub in-person scenario is higher than that of the other in-person scenarios in several impact categories, especially those contributed significantly by car transportation, including freshwater ecotoxicity, marine ecotoxicity, and terrestrial acidification.” — It appears that all multi-hub conference formats currently analyzed in the manuscript involve some attendees flying to conferences. Many short-haul flights (e.g. many US and European scientists flying a one to few thousand miles to a city in the same country/continent) can lead to very high CO₂ emissions (compared to scenarios where fewer academics fly longer distances to conferences). What would be the carbon footprint of hybrid conferences that involve no flying (i.e. local in-person hubs+virtually connected virtual conference: <https://www.nature.com/articles/d41586-019-03899-1>)? It would be valuable to include the carbon footprint analysis for this viable and promising hybrid format.

7. Lines 340-342, the authors state that: “However, the purpose of conferences to promote communication and collaboration would be violated if the daily participation of virtual conferences declines. Therefore, diminishing virtual participation should not be considered as a strategy to mitigate carbon footprint.” — There is no evidence that lower daily participation necessarily leads to lower scientific communication or collaboration. There could be many reasons for lower daily participation (e.g. an ongoing pandemic). There is no data to support that daily participation at virtual conferences is any different from the legacy in-person conferences where travel fatigue or tourism activities could disrupt attendee participation. In-person conferences involve jetlag fatigue and parallel scientific sessions tightly scheduled in 3-5 days. Virtual conferences enable more researchers from all career stages to attend (see references listed above) thus increasing the chances for interaction and communication. Their digital format provides contact information and recorded talks/papers/abstracts available to all during and after the conferences, creating more opportunities for exchange and follow-up discussion year-long.

8. Line 556 (Data availability): Please include all raw and analyzed data (from any source) used for analysis for this manuscript as Excel files in the supplementary information section. Apart from evaluating the analysis and conclusions made in this manuscript, the data can be valuable for readers/researchers who plan to continue and built upon this important work.

Reviewer #2 (Remarks to the Author):

The manuscript focuses on the carbon footprint of conferences. Since the conference business, in academics and elsewhere, has become a big industry with millions of persons attending events all around the world, understanding the impact of this behavior on climate change is clearly an important endeavor. The authors perform a full life cycle assessment of in person, hybrid and online

conferences. This comprehensive approach estimating carbon footprints and the cumulative energy demands is, as far as I know, very much unique to date.

As a major result the article shows that switching from an in person to a completely virtual conference reduces the carbon footprint by 94%. From a climate change perspective, online conferences are thus clearly the best option. However, the analysis also shows that hybrid solutions, with several optimally distributed conference locations and a proportion of online participation, also have considerable potential for saving emissions. According to the calculations, up to 70% are possible compared to an in-person conference at one location.

The paper includes in my view two major innovations compared to earlier studies: 1) it shows to what extent hybrid conferences at multiple hubs with a spatially optimized distribution of participants helps to reduce the environmental impact of conferences, and 2) it estimates not only the carbon footprint but applying a complete LCA approach the impact of conferences (virtual/hybrid and in person) on a number of outcomes (e.g. water depletion, or marine ecotoxicity). While the first innovation is made very clear, the second innovation is less prominent in the paper. To some extent I am also not sure whether focusing solely on the carbon footprint as the major outcome variable would be better in order not to overstretch the paper.

All in all, I think this is a very important article which helps to base the decision on how to move forward with the conference business on a stable empirical base. It therefore makes a significant contribution to the literature and fits well into the scope of Nature Communications. In the final section the authors could give more clear recommendations for conference organizers how to organize these events in an environmentally friendly manner. From a methodological point of view, I have to admit that I am definitely not an expert in LCA. The following questions should be considered accordingly against this background.

- The authors state that other studies on this topic mostly focused on the carbon footprint of travelling and neglected further emissions of in person and online conferences. While this point is true for most studies I am aware of, one study compares all three variants (in-person, online and hybrid) for the major European political science conference (Jäckle, 2021). Although clearly not as sophisticated in the methodological LCA approach as this paper, the results are nevertheless very similar. It also shows that hybrid solutions can – particularly if those participants who would have to fly in from far away join online – and all others accept longer travel times by bus/train instead of flying can significantly reduce the emissions from conferences.

- The authors use data from 2020 American Center for Life Cycle Assessment virtual conference – which is a medium sized conference with 536 participants, mostly from the US. The question of course is, to what extent this conference can be regarded as a representative event for (international) academic conferences in general. In addition to the maps in figure 3, the authors provide the breakdown of participants by country table S1. This shows that 88% are US or Canada based. Perhaps comparing this distribution to other international conferences would be a good idea. Knowing to what extent the results from this analysis are about the same or systematically different for other (international) conferences would help to strengthen the relevance of the results. E.g. it could be argued that conferences within Europe, where many participants can reach the venue via a high-speed train network, produce less emissions than within the US where no adequate train network exists and participants even from the US either fly or use their car.

- Concerning the impact from food and heating/electricity at the conference venue vs. participants staying at home. I was wondering whether the estimations assume that at home footprint from food and also from electricity/heating is the same as at the conference venue/conference hotel. In my view this would not make much sense since conference catering or eating at a restaurant probably produces much more food waste, than cooking at home. The same is probably true for heating/electricity: at a large conference venue/hotel I assume the footprint to be higher than at home. I did not find any information on this issue in the paper.

- Concerning the travel induced footprints the authors assume that participants take the train in Europe if the travel distance is below 600 km. I have two questions here: 1) is the travel distance for the train travel calculated by Google Distance Matrix API actually based on the railway network? Or is it based on a street network? Since in many European countries the railway network differs a lot from the street network in terms of density and directness (e.g. in France the high-speed train network connects all major cities with Paris in straight lines, whereas in Germany the train network follows the routes historically from one city to the next, which results often in less direct connections than using the street-network). 2) Assuming a certain threshold for traveling by car/bus/train or by airplane makes sense, yet I doubt that the travel distance itself is the right factor here. I suppose that most conference participants decide primarily on the basis of the travel time. Thus in cases where a good high-speed train network exists such as in France (e.g. travelling from Marseille to Paris is 775 km by car takes about 8h, by train less than 4h; flying is probably no real option here). Thus, using not a fixed km-threshold, but a threshold for the time accepted for getting to the conference venue might be a better option. See Jäckle (2019, 2021) for a similar approach.

- Different organizations report different emission factors for trains, cars and airplanes depending on various assumptions (European Environment Agency, 2021; UK Department for Business, Energy & Industrial Strategy, 2021; Umweltbundesamt, 2021). E.g. on the electricity mix used to power the trains, the average passenger-load factor, whether radiative forcing is included or not for flying etc. This naturally leads to deviating results in the estimation of the overall carbon footprint. The authors did not state – at least I did not find anything on this issue in the paper or the supplement – which assumptions they make, i.e. which emission factors they use for their estimates. Instead of presenting only point estimates it could also make sense to present a minimum-to-maximum range. This might particularly be the case for the travel induced carbon footprint since it clearly – as the results in the paper show – make up by far the highest percentage of emissions for the whole conference attendance. Variations in the emission factors are therefore very relevant for the overall estimates.

- Since air-travel is by far the biggest component of the carbon footprint of conferences, it makes sense to be as exact as possible here. I was wondering whether the distances used to calculate the carbon emissions from flying are solely the great circle distances between the airports or whether the fact that many airplanes do not fly the shortest route but more inefficient detours, or have stop overs has been included in the estimations. It has been shown that the actual distances aircraft fly are between 6 and 10% longer than the great circle routes between the departure and the destination airports (Kettunen et al., 2005). Furthermore, is there a difference in the estimation between long-haul and short-haul flights? Since long-haul flights reach higher altitudes where CO₂ exerts more harmful effects it may makes sense to make a distinction. Does the analysis account for the different electricity mix in the participants' countries for those joining virtually?

Minor point:

- Fig 2c is difficult to read. The colors/shading of the legend should be enlarged.

best regards,
Sebastian Jäckle

- European Environment Agency (2021) CO₂-emission intensity from electricity generation. Available at: <https://www.eea.europa.eu/data-and-maps/daviz/sds/co2-emission-intensity-from-electricity-generation-2/@@view>.

- Jäckle S (2019) WE have to change! The carbon footprint of ECPR general conferences and ways to reduce it. *European Political Science* 18(4): 630–650. DOI: 10.1057/s41304-019-00220-6.

- Jäckle S (2021) Reducing the Carbon Footprint of Academic Conferences by Online Participation: The Case of the 2020 Virtual European Consortium for Political Research General Conference. *PS: Political Science & Politics* 54(3). 2021/02/22 ed. Cambridge University Press: 456–461. DOI: 10.1017/S1049096521000020.

- Kettunen T, Hustache J-C, Fuller I, et al. (2005) Flight Efficiency Studies in Europe and the United States. In: 6th USA/Europe Seminar on ATM Research and Development, Baltimore, 27 June 2005. Available at: http://www.atmseminar.org/seminarContent/seminar6/papers/p_055_MPM.pdf.

- UK Department for Business, Energy & Industrial Strategy (2021) Greenhouse gas reporting: conversion factors 2020. Available at: <https://www.gov.uk/government/publications/greenhouse-gas-reporting-conversion-factors-2020>.

- Umweltbundesamt (2021) Bilanz 2019: CO₂-Emissionen pro Kilowattstunde Strom sinken weiter. Available at: <https://www.umweltbundesamt.de/presse/pressemitteilungen/bilanz-2019-co2-emissionen-pro-kilowattstunde-strom>.

Reviewer #3 (Remarks to the Author):

Dear authors, the paper is well-structured and covers an important gap in scientific literature on the GWP of virtual and hybrid conferences. However, before to be published some adjustments are still needed.

Introduction, I would extend the introduction or added a paragraph on the current state of the art on LCA and Carbon Footprint of conferences, to clearly highlight the innovative aspects of the current studies. You cited some studies with 4-13 references, I would add at least a table and a chapter indicating the main content in the previous papers.

The paper has lots of information, but it is not always clear reading it which are the data you have used and there are some Life cycle inventory data missing in the main manuscript. I know that the most are reported in the supplementary materials but it doesn't help the transparency of the study. Please report the main data of life cycle inventory in the main manuscript: e.g. the consumption of each transport system considered, the FAO data on Food, and so on.

The sequence of the manuscript is also not conformed to the an LCA study in according to the ISO

14040: Methodology with all 4 phases of LCA should be reported after the introduction and state of the art to allow the reader to understand the environmental impacts obtained. It is very important for an LCA study to be transparent in its assumptions made and ensuring transparency and reproducibility.

In page 7 Fig. 2 Comparison between the 1-hub in-person conference and the virtual conference. a, Carbon footprint associated with transportation for individual participants, indicating that the unit carbon footprint for trips primarily by plane tends to be smaller than that for driving. I do not see this result in the figure and it is not coherent with the results of other studies, please clarify it.

Trend of virtual and hybrid conferences since COVID-19 effectively mitigate climate change

Yanqiu Tao¹, Debbie Steckel², Fengqi You^{1,3*}

¹ Systems Engineering, Cornell University, Ithaca, New York, 14853, USA

² The American Center for Life Cycle Assessment, Bethesda, Maryland, 20824, USA

³ Robert Frederick Smith School of Chemical and Biomolecular Engineering, Cornell University, Ithaca, NY, 14853, USA

Response to reviewers and actions taken

Reviewer 1

The authors are most grateful to the editor for the helpful comments.

Reviewer's comment (1)

1. Lines 42-44, the authors write: “Whether future conferences should return in-person, keep purely virtual, or evolve to a hybrid of both has been widely debated (19).”— the phrase “kept purely virtual” could improve to “kept fully virtual” to better describe the conference organization mode.

Answer:

Thank you for your comments. We have improved this phrase in the revised manuscript, following the reviewer's suggestions.

Actions:

On page 3 of the revised manuscript:

“Whether future conferences should return in-person, keep fully virtual, or evolve to a hybrid of both has been widely debated¹⁹.”

References:

19 Moss, V. A. et al. Forging a path to a better normal for conferences and collaboration. Nature Astronomy 5, 213-216, (2021).

Reviewer's comment (2)

2. Lines 53-55, the authors write that: “By contrast, supporters of virtual conferences consider virtual interactions as a far more accessible, inclusive and sustainable counterpart to the traditional in-person format.” — a number of references that the authors may wish to use to support this statement (as the virtual conferences of 2020-2021 have improved equity and inclusivity in terms of gender, career stage, geography, disability status of attendees and speakers) are:

*<https://www.nature.com/articles/s41562-021-01067-y>,
[https://www.cell.com/trends/cognitive-sciences/fulltext/S1364-6613\(21\)00009-7?rss=yes](https://www.cell.com/trends/cognitive-sciences/fulltext/S1364-6613(21)00009-7?rss=yes),
<https://elifesciences.org/articles/62668> , <https://elifesciences.org/articles/57892>).*

Answer:

Thank you for your comments. We have added all these references to support this sentence in the revised manuscript, following the reviewer's suggestions.

Actions:

On page 3 of the revised manuscript:

“By contrast, supporters of virtual conferences consider virtual interactions as a far more accessible, inclusive, and sustainable counterpart to the traditional in-person format^{18,21-23}.”

References:

- 18 Sarabipour, S. et al. Changing scientific meetings for the better. *Nature Human Behaviour* 5, 296-300, (2021).
- 21 Achakulvisut, T. et al. Towards Democratizing and Automating Online Conferences: Lessons from the Neuromatch Conferences. *Trends in Cognitive Sciences* 25, 265-268, (2021).
- 22 Sarabipour, S. Virtual conferences raise standards for accessibility and interactions. *eLife* 9, e62668, (2020).
- 23 Achakulvisut, T. et al. Improving on legacy conferences by moving online. *eLife* 9, e57892, (2020).

Reviewer's comment (3)

3. Lines 55-56, the authors write that: “As a compromise of the two, hybrid conferences may be a viable solution.” — A successful hybrid conference that can be referenced here to support this statement was held in 2019: (<https://www.nature.com/articles/d41586-019-03899-1> , a permanent website for that event: <https://careconferences.org/>

Answer:

Thank you for your comments. We have cited this work to support this statement in the revised manuscript, following the reviewer's suggestions.

Actions:

On page 3 of the revised manuscript:

“As a compromise of the two, hybrid conferences may be a viable solution^{24,25}.”

References:

- 24 Abbott, A. Low-carbon, virtual science conference tries to recreate social buzz. 577, 13, (2019).
- 25 European Biological Rhythms Society. How does a CARE Conference work?, (2019) (Accessed 09/13/2021) <https://careconferences.org/>.

Reviewer's comment (4)

4. Lines 136-141, the authors state that: “Other than the food preparation, electricity consumption at home during the virtual conference contributes considerably to most impact categories. Notably, the environmental profile of electricity production is highly dependent on the geographical distribution of participants. The result indicates that the environmental benefits of virtual conferences are weakened if a large number of participants are located in regions without a strong penetration of renewables in the power grid.” — would the electricity consumption of attendees at virtual conferences be different from daily electricity consumption of academics in any location at work (daily electricity consumption of researchers working at their desktop or laptop computer remotely from home) or on campus/in office? Would the electricity consumption be different if most or some conference talks are watched in recorded format later (as opposed to live streaming)?

Answer:

Thank you for your comments. During the virtual conferences, both the electricity used for accommodation and the information and communication technology (ICT) service contribute to the electricity consumption. The electricity used for accommodation is estimated based on the average daily residential electricity consumption (U.S. Energy Information Administration, 2020); The electricity use for the ICT services involved with the video-conferencing is quantified following existing literature (Pärssinen et al., 2018) and is detailed in the Method section of the original manuscript. In terms of the electricity consumption for watching recordings after the virtual talk, the electricity consumption can be different from attending the real-time virtual talk. Both ways download a comparable amount of data while real-time virtual talk simultaneously uploads the video/audio of the participant. Moreover, recording can be played many times, and the electricity consumption depends on the storage method. If they are downloaded to the local hardware or solid-state drive, no internet connection is needed, but the embodied energy of these devices should be calculated in addition to the electricity use of the laptop/tablet/mobile during video-playing. If the video is shared through online platforms such as YouTube, then the electricity consumption accumulates with the number of plays. There lacks a reliable survey on the method and habit of sharing and playing recordings. Moreover, these post-conference activities are considered outside the system boundary of the life cycle assessment study in this work, because they are not part of the conference activities. Hence, we do not account for these consequential environmental impacts. We have clarified these issues in the revised manuscript to avoid confusion, following the reviewer's suggestions.

References:

Pärssinen, M., Kotila, M., Cuevas, R., Phansalkar, A. & Manner, J. Environmental impact assessment of online advertising. *Environmental Impact Assessment Review* 73, 177-200, (2018).

U.S. Energy Information Administration. State Energy Consumption Estimates 1960 Through 2018, Table C17, (June 2020) (Accessed 03/23/2021) <https://www.eia.gov/state/seds/archive/seds2018.pdf>.

Actions:

On page 8 of the revised manuscript:

“Other than the food preparation, electricity consumption for accommodation during the virtual conference and the video-conferencing contribute considerably to most impact categories.”

On page 19 of the revised manuscript:

“However, diminishing synchronous virtual participation may add difficulty to communication and collaboration for virtual conferences that do not provide sufficient asynchronous attendance options, such as the recording proceedings⁴². To create more opportunities for exchange and follow-up discussion, virtual conference organizers are encouraged to provide contact information, recording proceedings, and electronic documents submitted by the presenters to all participants. These asynchronous attendance practices could support virtual conferences to improve equity, diversity, inclusivity, networking, early career research promotion, and career development^{18,21-23}. On the other hand, post-conference activities, such as downloading and playing asynchronous recordings, sending follow-up emails, and searching for materials, can result in environmental impacts. The relationship between daily virtual participation and the amount of post-conference activities is unknown. Therefore, we cannot expand our system boundary to account for the variation in the environmental impacts of these consequential activities. Future work could further explore the consequential environmental impacts of the rapidly expanding video-conferencing industry.”

On page 20 of the revised manuscript:

“Post-conference activities of participants, such as downloading and playing asynchronous recordings, sending follow-up emails, and searching for materials, are out of the scope of this attributional LCA.”

On page 25 of the revised manuscript:

“*Information and communication technology*

To estimate the environmental impacts of virtual conferences, we incorporate an environmental impact assessment framework of Internet services⁵³. The system boundary of the ICT stage consists of infrastructure, network, and server related to video-conferencing. Specifically, four analysis layers are examined, including the energy consumption of the infrastructure, the shares of access network traffic and shares of IP protocols delivering the investigated Internet services, shares of traffic classes representing the end-user activities, and shares of the investigated Internet services in each traffic class. As participants of conferences usually use search engines or other office software simultaneously on the computer, mobile devices are excluded from the system boundary. Energy consumption of infrastructure, including router, Internet access equipment, computer, and data center equipment (cooling systems, lighting, and power supplies), are considered following previous studies^{39,54}. Additionally, the production and distribution of router, Internet access equipment, and computer are included in the system boundary, while the construction of data center infrastructure is excluded, because of the unavailable LCIA data and dominance of the operational phase to the overall environmental impacts⁵⁵. The energy intensity of the network for the video traffic class ($E_{Inetwork}$) is calculated as follows:

$$EI_{network} = \frac{\lambda_{fixed\ line} \cdot \lambda_{IPv4} \cdot \lambda_{TCP} \cdot \lambda_{HTTP} \cdot \lambda_{video} \cdot \sum_{k \in K} E_k}{DT_{video}} \quad (1)$$

where $\lambda_{fixed\ line}$, λ_{IPv4} , λ_{TCP} , λ_{HTTP} , λ_{video} represent the share of fixed line traffic in the total IP traffic, the share of IPv4 based traffic in the total IP traffic, the share of transmission control protocol (TCP) based traffic in the total IP traffic, the share of hypertext transfer protocol (HTTP) based traffic in the TCP based traffic, the share of video traffic class in the HTTP based traffic, respectively. K indicates the set of network components, including packet switched core, fixed line customer premises equipment, operator data center, office networks, and Internet core. E_k denotes the energy consumption of each component of the network. DT_{video} represents the fixed line traffic for the video traffic class. As the best available data of the network’s energy consumption and data traffic is from 2016, an annual electricity efficiency improvement of 10% is considered following the setting of the expected scenario in previous work⁴³. The energy intensity of the server (EI_{server}) is estimated from the best available data of a 2015 Sweden study on the ICT sector⁵⁶, and extrapolated to the value for 2020 with an annual electricity efficiency improvement of 10%⁴³. The calculation is as follows:

$$EI_{server} = \frac{E_{server}}{DT_{server}} \quad (2)$$

where E_{server} represents the total energy consumption of servers, and DT_{server} represents the total data traffic of the server. Data traffic of the virtual conference is computed following the survey of a recent study, which recognized 80% of participants attending the conference each day with a daily online duration of 5.5 hours⁵. The energy consumption related to the network and server for the virtual conference ($VE_{network}$ and VE_{server}) is computed following equation (3) and (4).

$$VE_{network} = EI_{network} \cdot DT_{virtual} \quad (3)$$

$$VE_{server} = EI_{server} \cdot DT_{virtual} \quad (4)$$

where $DT_{virtual}$ is the data traffic of the virtual conference calculated by multiplying the bandwidth of the video-conferencing with the total amount of online time for all participants. Downstream and upstream bandwidth of a Zoom group video calling for 720 high-definition videos is obtained from Zoom as 1.8 and 2.6 megabits per second (Mbps), respectively⁵⁷. Following the 2020 projection with expected improvement in energy efficiency from a previous study⁵⁴, the energy consumption of data center can be broken down into four equipment categories, namely infrastructure (33%), network (3%), storage (11%), and servers (53%). The material and energy inventories of the ICT stage are summarized in Table 6.”

Table 6 Material and energy consumption of the information and communication technology for virtual-conferencing^{5,39,43,53,54,56,57}.

Category	Item	Value	Unit
Device	Router, internet	1.09E-01	unit
	Internet access equipment	1.03E+00	unit
	Computer, laptop	1.54E+00	unit
Device-related electricity	Energy consumption for router	0.00E+00	kWh
	Energy consumption for internet access equipment (ADSL, DSLAM)	0.00E+00	kWh
	Energy consumption for laptop, video mode	3.43E+02	kWh
	PS-Core	2.38E+00	kWh
Network-related electricity	Fixed line CPE	3.83E+00	kWh
	Operator DC	4.75E-01	kWh
	Office networks	1.18E+00	kWh
	Internet core	2.88E-01	kWh
	Infrastructure	6.67E-01	kWh
Server-related electricity	Network	5.56E-02	kWh
	Storage	2.22E-01	kWh
	Server	1.07E+00	kWh

References:

- 5 Burtscher, L. et al. The carbon footprint of large astronomy meetings. *Nature Astronomy* 4, 823-825, (2020).
- 18 Sarabipour, S. et al. Changing scientific meetings for the better. *Nature Human Behaviour* 5, 296-300, (2021).
- 21 Achakulvisut, T. et al. Towards Democratizing and Automating Online Conferences: Lessons from the Neuromatch Conferences. *Trends in Cognitive Sciences* 25, 265-268, (2021).
- 22 Sarabipour, S. Virtual conferences raise standards for accessibility and interactions. *eLife* 9, e62668, (2020).
- 23 Achakulvisut, T. et al. Improving on legacy conferences by moving online. *eLife* 9, e57892, (2020).
- 39 Wernet, G. et al. The ecoinvent database version 3 (part I): overview and methodology. *Int. J. Life Cycle Assess.* 21, 1218-1230, (2016).
- 42 Fulcher, M. R. et al. Broadening Participation in Scientific Conferences during the Era of Social Distancing. *Trends Microbiol* 28, 949-952, (2020).
- 43 Andrae, A. S. G. & Edler, T. On Global Electricity Usage of Communication Technology: Trends to 2030. *Challenges* 6, (2015).
- 53 Pärssinen, M., Kotila, M., Cuevas, R., Phansalkar, A. & Manner, J. Environmental impact assessment of online advertising. *Environmental Impact Assessment Review* 73, 177-200, (2018).
- 54 Shehabi, A., Smith, S. J., Masanet, E. & Koomey, J. Data center growth in the United States: decoupling the demand for services from electricity use. *Environmental Research Letters* 13, 124030, (2018).

- 55 Whitehead, B., Andrews, D. & Shah, A. The life cycle assessment of a UK data centre. *The International Journal of Life Cycle Assessment* 20, 332-349, (2015).
- 56 Malmmodin, J. & Lundén, D. The energy and carbon footprint of the ICT and EaM sector in Sweden 1990-2015 and beyond. (2016).
- 57 Zoom Help Center. System requirements for Windows, macOS, and Linux (Accessed 03/23/2021) <https://support.zoom.us/hc/en-us/articles/201362023>.

Reviewer's comment (5)

5. The size of conferences (number of attendees) is an important consideration for conference organizers assessing features of meetings and optimizing the format and organization. This is highly variable (hundreds to tens of thousands) and an important factor for consideration during organization of the legacy(in-person) conferences. Have the authors performed sensitivity analysis on the impact of the number of attendees as a parameter under various scenarios with regards to conference format (in-person, hybrid, fully virtual) and their recommendations? It would be beneficial to include this to the manuscript text.

Answer:

Thank you for your comments. A larger number of attendees would increase the total environmental impact of the conference and vice versa. Since we report the environmental impact of the conference on a per capita basis, the number of attendees alone would not alter the results of in-person, hybrid, or fully virtual scenarios. However, the geographic distribution of the attendees may impact the per capita environmental sustainability of the conference. Instead of the sensitivity analysis on the distribution of participants, we have added a Comparison with the state-of-the-art conference LCA studies subsection in the revised manuscript to summarize the range of carbon footprint results obtained from existing studies, following the reviewer's suggestion.

Actions:

On page 6 of the revised manuscript:

“Comparison with the state-of-the-art conference LCA studies

Supplementary Table S1 summarizes and compares the state-of-the-art conference LCA studies. Previous LCA studies mainly focused on quantifying the carbon footprint of in-person conferences, while only two of them presented results for other impact categories using the life cycle impact assessment (LCIA) method, including CML2001, USEtox, Eco-Indicator 99, and UBP 97^{6,13}. Half of them focused exclusively on round-trip transportation^{4,5,9,11,12,29} while the rest considered life cycle stages of preparation, execution, catering, accommodation, and transportation^{6-8,13,26}. However, due to differences in assumptions associated with in-person conferences (e.g., duration, size, and locations of the conference, geographical distribution of participants, transportation mode, system boundary, and selection of characterization factors), the carbon footprint ranges from 92 to 3540 kg CO₂ eq. per capita. All of these studies identified transportation as the environmental hotspot. The conference site and geographical distribution of participants determine the transportation distance and mode for participants. From those who reported the average transportation distance, the average round-trip transportation distance varies from 1980 km¹³ to 9564 km⁹. However, the average distance does not illustrate the complete picture of the participant transportation, and it was found that the 10–20% of participants with the most polluting trips contribute to a substantial portion (20–70%) of the total transportation-induced emissions^{4-7,12}. These values depend on the distribution of participants, which reveals whether the conference is more localized or more internationalized. As shown in Supplementary Table S1, most participants are from the region where the conferences are held. The conference location is also important in

determining the transportation profile, in which a conference location with better train connection to other major cities is capable of allowing more participants to transport by train and thus has more potential of reducing carbon footprint while a conference located in the southern hemisphere usually perform much worse in terms of carbon footprint compared to the northern hemisphere^{11,12}. Bossdorf et al. suggested food and accommodation accounted for 18% and 13% of the total carbon footprint, respectively⁸. On the other hand, owing to the exclusive vegetarian menu and the much higher transportation emissions, Astudillo and Azarijafari reported that food and accommodation only accounted for 1% and 2% of the total carbon footprint⁷.

Recent studies compared the carbon footprint of in-person and virtual conferences^{4,5,6}, which ranges from 0 to 5.87 kg CO₂ eq. per capita. Specifically, Ewijk and Hoekman assumed carbon neutrality for the virtual conference⁴; Jäckle computed the carbon footprint from the electricity needed for devices and servers²⁶; Burtscher et al. considered emissions related to network, laptop, and Zoom-server⁵. Several studies considered multi-site conferences, yet the choices of locations are arbitrary^{4,13}. Stroud and Feeley chose to optimize the conference location by minimizing the carbon footprint while restricting the potential locations to participants' origins⁹. Astudillo and Azarijafari considered the geometric median of all participants as the optimal conference location⁷. As discussed above, none of the existing studies explicitly explored to what extent virtual conferences and multi-hub hybrid conferences with spatially optimized conference hubs and participant assignments can reduce the environmental impacts of in-person conferences.”

On page 18 of the revised manuscript:

“As summarized in Supplementary Fig. S1, the carbon footprint of the 1-hub in-person conference reported in this work (840 kg CO₂ eq. per participant) is consistent with the previous LCA results on in-person conferences⁴⁻¹³, and is comparable to the monthly carbon footprint of an average U.S. citizen in 2018⁴¹.”

On page 18 of the revised manuscript:

“As we consider the whole life cycle of the virtual conference, less reduction of carbon footprint can be achieved by switching to full virtual or hybrid mode, compared to values reported by other studies^{4,5,13,26}.”

References:

- 4 van Ewijk, S. & Hoekman, P. Emission reduction potentials for academic conference travel *Journal of Industrial Ecology* n/a, (2020).
- 5 Burtscher, L. et al. The carbon footprint of large astronomy meetings. *Nature Astronomy* 4, 823-825, (2020).
- 6 Neugebauer, S., Bolz, M., Mankaa, R. & Traverso, M. How sustainable are sustainability conferences? – comprehensive Life Cycle Assessment of an International Conference Series in Europe. *Journal of Cleaner Production* 242, 118516, (2019).
- 7 Astudillo, M. F. & Azarijafari, H. Estimating the global warming emissions of the LCAXVII conference: connecting flights matter. *The International Journal of Life Cycle Assessment* 23, 1512-1516, (2018).
- 8 Bossdorf, O., Parepa, M. & Fischer, M. Climate-neutral ecology conferences: just do it! *Trends in ecology & evolution* 25, 61, (2009).
- 9 Stroud, J. T. & Feeley, K. J. Responsible academia: optimizing conference locations to minimize greenhouse gas emissions. *Ecography* 38, 402-404, (2015).
- 11 Spinellis, D. & Louridas, P. The Carbon Footprint of Conference Papers. *PLOS ONE* 8, e66508, (2013).
- 12 Desiere, S. The Carbon Footprint of Academic Conferences: Evidence from the 14th

- EAAE Congress in Slovenia. EuroChoices 15, 56-61, (2016).
- 13 Hischier, R. & Hilty, L. Environmental impacts of an international conference. Environmental Impact Assessment Review - ENVIRON IMPACT ASSESS REV 22, 543-557, (2002).
- 26 Jäckle, S. Reducing the Carbon Footprint of Academic Conferences by Online Participation: The Case of the 2020 Virtual European Consortium for Political Research General Conference. PS: Political Science & Politics 54, 456-461, (2021).
- 29 Jäckle, S. WE have to change! The carbon footprint of ECPR general conferences and ways to reduce it. European Political Science 18, 630-650, (2019).
- 41 IEA. CO2 Emissions from Fuel Combustion, (2018) (Accessed 04/19/2021) <http://energyatlas.iea.org/>.

On page 2 of the revised Supplementary Information:

“

Supplementary Fig. S1 Comparison of per capita carbon footprint results from previous literature and this study. By taking the whole life cycle of a virtual conference into consideration, the carbon footprint of the virtual conference in this study is substantially higher than those in

other studies¹⁻⁴. The carbon footprint of 1-hub in-person conferences is within the range of values reported by existing studies. Only a few studies investigated the carbon footprint of multi-hub in-person conferences. The carbon footprint of 1-hub and 2-hub in-person conferences from Ewijk and Hoekman³ is higher than that from our study because the participants of that conference are more geographically distributed than those in this study. Due to the same reason, the carbon footprint of the 3-hub in-person conference from Ewijk and Hoekman³ is significantly reduced and becomes lower than our result.

Supplementary Table S1 Literature comparison on the life cycle assessment (LCA) of conferences. Abbreviations: GWP, global warming potential; ICT, information and communication technology; CED, cumulative energy demand; LCIA, life cycle impact assessment; NH, New Hampshire; US, the United States; UK, the United Kingdom.

Author, Year	Conference mode	Number of conference hubs	Number of participants	Geographical distribution of participants	Duration (Day)	Location	GWP per capita (kg CO ₂ eq)	Life cycle stages	LCIA method
Jäckle, 2021 ¹	In-person, virtual, hybrid	1	2208	-	5	Innsbruck, Austria	In-person: 566-1166 Virtual: 0.35 (0.03-5.87)	Catering, accommodation, transportation	GWP
Burtscher et al., 2020 ²	In-person, virtual	0,1	1240 (in-person) 1777 (virtual)	84% from Europe (in-person)	5	Lyon, France Virtual	In-person: 1500 Virtual: 0.33	Transportation ICT	GWP
Ewijk and Hoekman, 2020 ³	In-person, virtual, hybrid (1 hub, 10% virtual)	0,1,2,3	625 (in-person) 588 (in-person) 401 (in-person)	Dependent on conference site*	-	Chicago, US Surrey, UK Ulsan, South Korea Virtual	In-person (1 hub): 1513 (1240-1830) In-person (2 hubs): 770 In-person (3 hubs): 450 (330-500) Hybrid: 920-1440 0 (virtual)	Transportation	GWP
Jäckle, 2019 ⁵	In-person, hybrid	1	1930	Europe: 89% North America: 4% Asia: 4% Rest of World: 3%	3-4	Bordeaux, France Glasgow, UK Montreal, Canada Prague, Czechia Oslo, Norway Hamburg, Germany	500-3400	Transportation	GWP
Neugebauer et al., 2019 ⁶	In-person	1	800	Europe: 85% Asia: 5% North America: 4% South America: 3% Rest of World: 3%	3	Aachen, Germany	518-570	Preparation, execution, catering, accommodation, transportation	CML2001, USEtox
Astudillo and Azarijafari, 2018 ⁷	In-person	1	228	North America: 87% Europe: 5% Asia: 4% Rest of World: 4%	3	Portsmouth, NH, US	952	Transportation, catering, accommodation	GWP
Desiere, 2015 ⁸	In-person	1	646	Europe: 85% US: 4% Japan: 2%	-	Ljubljana, Slovenia	150-498	Transportation	GWP

* Most participants are from the continent where the conference is held

				Rest of World: 9%					
Stroud and Feeley, 2014 ⁹	In-person	1	311	-	-	Canary Island, Spain	2580-3220	Transportation	GWP
			207			Merida, Mexico	3000-3540		
			385			Crete, Greece	2510-3020		
			406			Miami, FL, US	2560-2910		
Spinellis and Louridas, 2013 ^{10†}	In-person	-	450	-	-	Bayreuth, Germany	2540-2720	Transportation	GWP
			-			Multi-location multi-year	801 (48 – 2396)		
Bossdorf et al., 2009 ¹¹	In-person	1	125	-	3	Bern, Switzerland	92	Preparation, execution, catering, accommodation, transportation	GWP
Hischier and Hilty, 2002 ⁴	In-person, virtual	0,1,3	308 (in-person) 1000 (virtual)	-	3	Zurich, Switzerland	In-person: 240 Virtual: 2.5 Hybrid: 137	Organization, materials, and transportation	Eco-Indicator 99 UBP 97 CO ₂
This study	In-person, virtual, hybrid	0,1,2,3,4,5,6	536	North America: 88% Europe: 8% Rest of World: 4%		Optimal location	55 (virtual) 455-840 (in-person) 64-1804 (hybrid)	Preparation, execution, catering, accommodation, transportation ICT	GWP CED ReCiPe

”

† This paper randomly sampled proceeding conference papers from Scopus

References:

- 1 Jäckle, S. Reducing the Carbon Footprint of Academic Conferences by Online Participation: The Case of the 2020 Virtual European Consortium for Political Research General Conference. *PS: Political Science & Politics* 54, 456-461, (2021).
- 2 Burtscher, L. et al. The carbon footprint of large astronomy meetings. *Nature Astronomy* 4, 823-825, (2020).
- 3 van Ewijk, S. & Hoekman, P. Emission reduction potentials for academic conference travel *Journal of Industrial Ecology* n/a, (2020).
- 4 Hischer, R. & Hilty, L. Environmental impacts of an international conference. *Environmental Impact Assessment Review - ENVIRON IMPACT ASSESS REV* 22, 543-557, (2002).
- 5 Jäckle, S. WE have to change! The carbon footprint of ECPR general conferences and ways to reduce it. *European Political Science* 18, 630-650, (2019).
- 6 Neugebauer, S., Bolz, M., Mankaa, R. & Traverso, M. How sustainable are sustainability conferences? – comprehensive Life Cycle Assessment of an International Conference Series in Europe. *Journal of Cleaner Production* 242, 118516, (2019).
- 7 Astudillo, M. F. & AzariJafari, H. Estimating the global warming emissions of the LCAXVII conference: connecting flights matter. *The International Journal of Life Cycle Assessment* 23, 1512-1516, (2018).
- 8 Desiere, S. The Carbon Footprint of Academic Conferences: Evidence from the 14th EAAE Congress in Slovenia. *EuroChoices* 15, 56-61, (2016).
- 9 Stroud, J. T. & Feeley, K. J. Responsible academia: optimizing conference locations to minimize greenhouse gas emissions. *Ecography* 38, 402-404, (2015).
- 10 Spinellis, D. & Louridas, P. The Carbon Footprint of Conference Papers. *PLOS ONE* 8, e66508, (2013).
- 11 Bossdorf, O., Parepa, M. & Fischer, M. Climate-neutral ecology conferences: just do it! *Trends in ecology & evolution* 25, 61, (2009).

Reviewer's comment (6)

6. Lines 229-231, the authors state that: “Adding more hubs leads to reductions in the air transportation distances, as shown in Fig. 4c. Therefore, impact categories led by air transportation achieve the most reduction benefits from the addition of conference hubs.” and in lines 318-322: “Adding more conference hubs lowers the average air transportation distance and increases the average travel distance by car at a decreasing rate. Due to the increasing travel distances by car, the environmental impact of the 6-hub in-person scenario is higher than that of the other in-person scenarios in several impact categories, especially those contributed significantly by car transportation, including freshwater ecotoxicity, marine ecotoxicity, and terrestrial acidification.” — It appears that all multi-hub conference formats currently analyzed in the manuscript involve some attendees flying to conferences. Many short-haul flights (e.g. many US and European scientists flying a one to few thousand miles to a city in the same country/continent) can lead to very high CO₂ emissions (compared to scenarios where fewer academics fly longer distances to conferences). What would be the carbon footprint of hybrid conferences that involve no flying (i.e. local in-person hubs+virtually connected virtual conference: <https://www.nature.com/articles/d41586-019-03899-1>)? It would be valuable to include the carbon footprint analysis for this viable and promising hybrid format.

Answer:

Thank you for your comments. We agree that short-haul flights have higher greenhouse gas emissions than those of medium- and long-haul flights on a per passenger-kilometer basis. Indeed, we took this factor into consideration and adopted the characteristic factors of air transportation that decrease over distance, as shown in Supplementary Fig. S2 of the original Supplementary Information.

In terms of the hybrid conferences that involve no flying, several “maximum virtual participation” hybrid scenarios, as summarized in Table 1, involve no flying due to the 600-km threshold for rail transport and 500-km threshold for road trips. Fig. 5a of the original manuscript shows that raising the virtual participation level to 70% can cut the carbon footprint of in-person conferences by over 80% for all scenarios with the same number of hubs. Following the reviewer’s suggestion, we have clarified this point in the revised manuscript.

Table 1. “Maximum virtual participation” hybrid scenarios with participants traveled by rail or car. Each cell corresponds to a scenario. For each scenario, the row represents the number of conference hubs, and the column represents the maximum virtual participation level. Cells highlighted in yellow indicates that no participants of the corresponding scenario travel by plane.

maximum virtual participation level	1 hub	2 hubs	3 hubs	4 hubs	5 hubs	6 hubs
10%						
30%						
50%						
70%						

On page 7 of the original Supplementary Information:

“

Supplementary Fig. S3 The carbon footprint and CED per passenger-km (pkm) of flight. We fit the characterization factors of air transportation over distance from Cox et al.³ to polynomial functions for the year 2020. The values of power, slope, intercept, and R^2 are summarized in Supplementary Table S3.

Supplementary Table S3 Parameters of the fitted power functions for the characterization factors of air transportation over distance from Cox et al.³.

Impact category	Power	Slope	Intercept	R^2
Global Warming Potential (kg CO ₂ eq per passenger km)	-8.39E-01	4.06E+01	1.12E-01	9.97E-01
Cumulative Energy Demand, Non-Renewable (MJ per passenger km)	-8.82E-01	6.08E+02	1.05E+00	9.98E-01
Agricultural Land Occupation Potential (m ² yr per passenger km)	-8.94E-01	4.29E-01	2.52E-04	9.99E-01
Fossil Depletion Potential (kg oil eq per passenger km)	-8.80E-01	1.39E+01	2.48E-02	9.98E-01
Freshwater Ecotoxicity Potential (kg 14-DCB eq per passenger km)	-9.19E-01	1.08E-01	8.57E-05	9.99E-01
Freshwater Eutrophication Potential (kg P eq per passenger km)	-9.57E-01	3.43E-03	1.30E-06	1.00E+00
Human Toxicity Potential (kg 1,4 DB eq per passenger km)	-9.83E-01	5.92E+00	2.07E-03	1.00E+00
Ionising Radiation Potential (kg U235 eq per passenger km)	-9.02E-01	3.67E+00	4.79E-03	9.99E-01
Marine Ecotoxicity Potential (kg 14-DCB eq per passenger km)	-9.30E-01	1.06E-01	6.63E-05	9.99E-01
Marine Eutrophication Potential (kg N eq per passenger km)	-9.20E-01	6.03E-02	1.38E-04	1.00E+00
Metal Depletion Potential (kg Fe eq per passenger km)	-9.15E-01	3.58E-01	3.45E-04	9.99E-01
Natural Land Transformation Potential (m ² per passenger km)	-8.81E-01	1.48E-02	2.60E-05	9.98E-01
Ozone Depletion Potential (kg CFC11 eq per passenger km)	-8.78E-01	7.00E-06	1.26E-08	9.98E-01

Particulate Matter Formation Potential (kg PM10 eq per passenger km)	-9.36E-01	4.31E-02	2.75E-05	1.00E+00
Photochemical Oxidant Formation Potential (kg NMVOC per passenger km)	-9.58E-01	1.65E-01	6.36E-05	1.00E+00
Terrestrial Acidification Potential (kg SO ₂ eq per passenger km)	-9.57E-01	2.32E-03	1.15E-06	1.00E+00
Terrestrial Ecotoxicity Potential (kg 14-DCB eq per passenger km)	-9.57E-01	2.32E-03	1.15E-06	1.00E+00
Urban Land Occupation Potential (m ² yr per passenger km)	-9.66E-01	3.78E-01	1.30E-04	1.00E+00
Water Depletion Potential (m ³ H ₂ O per passenger km)	-9.43E-01	1.17E-01	7.29E-05	1.00E+00

On page 14 of the original manuscript:

“

Fig. 5 Trade-offs between face-to-face communication and carbon footprint. Different markers indicate the results of different scenarios, and their colors refer to the number of conference hubs. Doughnut charts represent the breakdowns of scenarios, pointed by the blue dashed lines. **a**, Comparison between the hybrid scenarios with a constraint of the maximum virtual participation, the in-person scenarios, and the baseline virtual scenario. **b**, Comparison between

the hybrid scenarios with a constraint of the maximum travel distance, the in-person scenarios, and the baseline virtual scenario.”

References:

- 3 Cox, B., Jemiolo, W. & Mutel, C. Life cycle assessment of air transportation and the Swiss commercial air transport fleet. *Transportation Research Part D: Transport and Environment* 58, 1-13, (2018).

Actions:

On page 15 of the revised manuscript:

“For multi-hub “maximum virtual participation” scenarios, raising the virtual participation level to 70% eliminates air travel, resulting in over 80% reduction in the carbon footprint.”

Reviewer’s comment (7)

7. Lines 340-342, the authors state that: “However, the purpose of conferences to promote communication and collaboration would be violated if the daily participation of virtual conferences declines. Therefore, diminishing virtual participation should not be considered as a strategy to mitigate carbon footprint.” — There is no evidence that lower daily participation necessarily leads to lower scientific communication or collaboration. There could be many reasons for lower daily participation (e.g. an ongoing pandemic). There is no data to support that daily participation at virtual conferences is any different from the legacy in-person conferences where travel fatigue or tourism activities could disrupt attendee participation. In-person conferences involve jetlag fatigue and parallel scientific sessions tightly scheduled in 3-5 days. Virtual conferences enable more researchers from all career stages to attend (see references listed above) thus increasing the chances for interaction and communication. Their digital format provides contact information and recorded talks/papers/abstracts available to all during and after the conferences, creating more opportunities for exchange and follow-up discussion year-long.

Answer:

Thank you for your comments. This statement attempted to interpret the sensitivity analysis result on the parameter of daily virtual participation. We agree that diminished virtual participation may not lead to less scientific communication or collaboration if contact information, recordings, and electronic documents are available. However, some conferences do not provide asynchronous attendance options, such as contact information, electronic documents submitted by the presenters, and recorded proceedings (Fulcher et al., 2020). Such a situation leads to challenges in post-conference communication and discussion. If these asynchronous attendance options are provided, they can support virtual conferences to improve equity, diversity, inclusivity, networking, early career research promotion, and career development, as suggested by the references provided in Reviewer’s comment (2) (Sarabipour et al., 2021; Achakulvisut et al., 2021; Sarabipour 2020; Achakulvisut et al., 2020). Moreover, these asynchronous attendance options may result in additional environmental impacts due to internet access and information and communication technology (ICT) equipment usage (e.g., follow-up emails and video-conferencing, file downloading, material searching, and playing videos). Yet, there lacks reliable data on the relationship between the daily virtual participation and the amount of these post-conference activities. Hence, we do not account for these consequential environmental impacts. We have revised this statement to avoid confusion in the revised manuscript, following the reviewer’s suggestions.

References:

Fulcher, M. R. et al. Broadening Participation in Scientific Conferences during the Era of Social Distancing. Trends Microbiol 28, 949-952, (2020).

Sarabipour, S. et al. Changing scientific meetings for the better. Nature Human Behaviour 5, 296-300, (2021).

Achakulvisut, T. et al. Towards Democratizing and Automating Online Conferences: Lessons from the Neuromatch Conferences. Trends in Cognitive Sciences 25, 265-268, (2021).

Sarabipour, S. Virtual conferences raise standards for accessibility and interactions. eLife 9, e62668, (2020).

Achakulvisut, T. et al. Improving on legacy conferences by moving online. eLife 9, e57892, (2020).

Actions:

On page 19 of the revised manuscript:

“However, diminishing synchronous virtual participation may add difficulty to communication and collaboration for virtual conferences that do not provide sufficient asynchronous attendance options, such as the recording proceedings⁴². To create more opportunities for exchange and follow-up discussion, virtual conference organizers are encouraged to provide contact information, recording proceedings, and electronic documents submitted by the presenters to all participants. These asynchronous attendance practices could support virtual conferences to improve equity, diversity, inclusivity, networking, early career research promotion, and career development^{18,21-23}. On the other hand, post-conference activities, such as downloading and playing asynchronous recordings, sending follow-up emails, and searching for materials, can result in environmental impacts. The relationship between daily virtual participation and the amount of post-conference activities is unknown. Therefore, we cannot expand our system boundary to account for the variation in the environmental impacts of these consequential activities. Future work could further explore the consequential environmental impacts of the rapidly expanding video-conferencing industry. Dietary type as well as the electricity and food consumption rate are the next most sensitive variable for a virtual conference.”

References:

- 18 Sarabipour, S. et al. Changing scientific meetings for the better. Nature Human Behaviour 5, 296-300, (2021).
- 21 Achakulvisut, T. et al. Towards Democratizing and Automating Online Conferences: Lessons from the Neuromatch Conferences. Trends in Cognitive Sciences 25, 265-268, (2021).
- 22 Sarabipour, S. Virtual conferences raise standards for accessibility and interactions. eLife 9, e62668, (2020).
- 23 Achakulvisut, T. et al. Improving on legacy conferences by moving online. eLife 9, e57892, (2020).
- 42 Fulcher, M. R. et al. Broadening Participation in Scientific Conferences during the Era of Social Distancing. Trends Microbiol 28, 949-952, (2020).

Reviewer’s comment (8)

8. Line 556 (Data availability): Please include all raw and analyzed data (from any source) used for analysis for this manuscript as Excel files in the supplementary information section. Apart from evaluating the analysis and conclusions made in this manuscript, the data can be valuable for readers/researchers who plan to continue and built upon this important work.

Answer:

Thank you for your comments. All the raw and analyzed data has been provided as Excel files and available for download from GitHub (https://github.com/PEESEgroup/Virtual_Con), following the reviewer's suggestions.

Actions:

On page 30 of the revised manuscript:

“All data needed to evaluate the conclusions in the paper are present in the paper or can be accessed through the Supplementary Materials and GitHub (https://github.com/PEESEgroup/Virtual_Con) with the published version archived (<https://doi.org/10.5281/zenodo.5515049>).”

Trend of virtual and hybrid conferences since COVID-19 effectively mitigate climate change

Yanqiu Tao¹, Debbie Steckel², Fengqi You^{1,3*}

¹ Systems Engineering, Cornell University, Ithaca, New York, 14853, USA

² The American Center for Life Cycle Assessment, Bethesda, Maryland, 20824, USA

³ Robert Frederick Smith School of Chemical and Biomolecular Engineering, Cornell University, Ithaca, NY, 14853, USA

Response to reviewers and actions taken

Reviewer 2

The authors are most grateful to the editor for the helpful comments.

Reviewer's comment (1)

The paper includes in my view two major innovations compared to earlier studies: 1) it shows to what extent hybrid conferences at multiple hubs with a spatially optimized distribution of participants helps to reduce the environmental impact of conferences, and 2) it estimates not only the carbon footprint but applying a complete LCA approach the impact of conferences (virtual/hybrid and in person) on a number of outcomes (e.g. water depletion, or marine ecotoxicity). While the first innovation is made very clear, the second innovation is less prominent in the paper. To some extent I am also not sure whether focusing solely on the carbon footprint as the major outcome variable would be better in order not to overstretch the paper.

All in all, I think this is a very important article which helps to base the decision on how to move forward with the conference business on a stable empirical base. It therefore makes a significant contribution to the literature and fits well into the scope of Nature Communications. In the final section the authors could give more clear recommendations for conference organizers how to organize these events in an environmentally friendly manner. From a methodological point of view, I have to admit that I am definitely not an expert in LCA. The following questions should be considered accordingly against this background.

Answer:

Thank you for your comments. In this study, we account for full-spectral impact categories because some of the investigated life cycle stages have been reported contributing to many environmental issues, such as food consumption to land use, water use, and pollution of aquatic and terrestrial ecosystems (Springmann et al., 2018). Moreover, since the environmental profile of electricity consumption depends on the energy sources of the local power grid, it can be diverse across scenarios (United States Environmental Protection Agency, 2021). Thus, we believe there is a need to take environmental issues other than carbon footprint into consideration. Following the reviewer's comments, the second innovation has been made more prominent in the revised manuscript. Specifically, two figures have been added to the revised Supplementary Information

to compare impact categories other than the carbon footprint (i.e., cumulative energy demand (CED) and 17 ReCiPe midpoint indicators) for the virtual, in-person, and hybrid scenarios. The results for other impact categories show similar trends as the carbon footprint results. Therefore, we do not discuss them in detail in the revised manuscript. For conference organizers, more clear recommendations on sustainable practices, such as the trade-off between environmental sustainability and in-person participation level, the trade-off between environmental sustainability and organizational challenges (e.g., increasing number of conference hubs), dietary type, and resource consumption, has been added to the revised manuscript.

References:

Springmann, M. et al. Options for keeping the food system within environmental limits. *Nature* 562, 519-525, (2018).

United States Environmental Protection Agency. Emissions & Generation Resource Integrated Database (Accessed 01/22/2021) <https://www.epa.gov/egrid>.

Actions:

On page 16 of the revised manuscript:

“The results for other impact categories, including CED and 17 ReCiPe midpoint indicators, generally follow similar trends as the carbon footprint results. Details are shown in Supplementary Fig. S13-S14.”

On page 18 of the revised manuscript:

“Therefore, from the environmental perspective, it is beneficial to hold a hybrid conference and decide the location of the hubs using the registration information or survey responses. The hub locations can be sub-optimal because differences may exist between the pre-conference information and the real attendance. Adding hubs and increasing virtual participation levels tend to provide more environmental benefits, but this benefit becomes less prominent as the number of hubs and virtual participation level are high enough. It is therefore important for conference organizers to consider the trade-off between organizational challenges and environmental sustainability.”

On page 20 of the revised manuscript:

“For virtual conference organizers, advocating energy saving from heating and other residential electricity use (e.g., air conditioning, lighting, electronics, and appliances), food waste reduction, ovo vegetarian diet can be effective practices to improve sustainability.”

On page 17 of the revised Supplementary Information:

“

Supplementary Fig. S13 Carbon footprint, CED, and 17 ReCiPe midpoint indicators of the virtual, in-person, and “maximum travel distance” scenarios with 1 to 6 hubs for an average participant. The y axis lists the mode, specific constraint (i.e., maximum travel distance), and the number of conference hubs for each scenario. From top to bottom, scenarios with the same amount of conference hubs are ranked by their virtual participation level in descending order. The percentages are computed based on the environmental impacts of the 1-hub in-person conference scenario.”

On page 18 of the revised Supplementary Information:

“

Supplementary Fig. S14 Carbon footprint, CED, and 17 ReCiPe midpoint indicators of the virtual, in-person, and “maximum virtual participation” scenarios with 1 to 6 hubs for an average participant. The y axis lists the mode, specific constraint (i.e., maximum virtual participation), and the number of conference hubs for each scenario. From top to bottom, scenarios with the same amount of conference hubs are ranked by their virtual participation level in descending order. The percentages are computed based on the environmental impacts of the 1-hub in-person conference scenario.”

Reviewer’s comment (2)

- The authors state that other studies on this topic mostly focused on the carbon footprint of travelling and neglected further emissions of in person and online conferences. While this point is true for most studies I am aware of, one study compares all three variants (in-person, online and

hybrid) for the major European political science conference (Jäckle, 2021). Although clearly not as sophisticated in the methodological LCA approach as this paper, the results are nevertheless very similar. It also shows that hybrid solutions can – particularly if those participants who would have to fly in from far away join online – and all others accept longer travel times by bus/train instead of flying can significantly reduce the emissions from conferences.

Answer:

Thank you for your comments. We have discussed the methodology of this study in the Introduction section of the revised manuscript, following the reviewer’s suggestions.

Actions:

On page 4 of the revised manuscript:

“A recent study on the carbon footprint of virtual, in-person, and hybrid conferences accounted for the video-conferencing-related emissions, transportation, execution, catering, and accommodation²⁶. However, it considered a single conference hub for both in-person and hybrid conferences and thus neglected the geographical effects of hub selection and participant assignment.”

References:

26 Jäckle, S. Reducing the Carbon Footprint of Academic Conferences by Online Participation: The Case of the 2020 Virtual European Consortium for Political Research General Conference. *PS: Political Science & Politics* 54, 456-461, (2021).

Reviewer’s comment (3)

- The authors use data from 2020 American Center for Life Cycle Assessment virtual conference – which is a medium sized conference with 536 participants, mostly from the US. The question of course is, to what extent this conference can be regarded as a representative event for (international) academic conferences in general. In addition to the maps in figure 3, the authors provide the breakdown of participants by country table S1. This shows that 88% are US or Canada based. Perhaps comparing this distribution to other international conferences would be a good idea. Knowing to what extent the results from this analysis are about the same or systematically different for other (international) conferences would help to strengthen the relevance of the results. E.g. it could be argued that conferences within Europe, where many participants can reach the venue via a high-speed train network, produce less emissions than within the US where no adequate train network exists and participants even from the US either fly or use their car.

Answer:

Thank you for your comments. We agree that the ACLCA conference is a medium-size conference with most participants from North America, but since participants’ information from other conferences is usually not accessible, information from the ACLCA conference serves as the best available data for us to conduct a case study on the potential of conferences for reducing environmental impacts. Following the reviewer’s suggestions, we added a Comparison with the state-of-the-art conference LCA studies subsection in the revised manuscript to compare the participants’ distribution among those of other international conferences from previous studies and discuss the differences in their LCA results.

Actions:

On page 6 of the revised manuscript:

“Comparison with the state-of-the-art conference LCA studies

Supplementary Table S1 summarizes and compares the state-of-the-art conference LCA studies. Previous LCA studies mainly focused on quantifying the carbon footprint of in-person conferences,

while only two of them presented results for other impact categories using the life cycle impact assessment (LCIA) method, including CML2001, USEtox, Eco-Indicator 99, and UBP 97^{6,13}. Half of them focused exclusively on round-trip transportation^{4,5,9,11,12,29} while the other half considered life cycle stages of preparation, execution, catering, accommodation, and transportation^{6-8,13,26}. However, due to differences in assumptions associated with in-person conferences (e.g., duration, size, and locations of the conference, geographical distribution of participants, transportation mode, system boundary, and selection of characterization factors), the carbon footprint ranges from 92 to 3540 kg CO₂ eq. per capita. All of these studies identified transportation as the environmental hotspot. The conference site and geographical distribution of participants determine the transportation distance and mode for participants. From those who reported the average transportation distance, the average round-trip transportation distance varies from 1980 km¹³ to 9564 km⁹. However, the average distance does not illustrate the complete picture of the participant transportation, and it was found that the 10–20% of participants with the most polluting trips contribute to a substantial portion (20–70%) of the total transportation-induced emissions^{4-7,12}. These values depend on the distribution of participants, which reveals whether the conference is more localized or more internationalized. As shown in Supplementary Table S1, most participants are from the region where the conferences are held. The conference location is also important in determining the transportation profile, in which a conference location with better train connection to other major cities is capable of allowing more participants to transport by train and thus has more potential of reducing carbon footprint while a conference located in the southern hemisphere usually perform much worse in terms of carbon footprint compared to the northern hemisphere^{11,12}. Bossdorf et al. suggested food and accommodation accounted for 18% and 13% of the total carbon footprint, respectively⁸. On the other hand, owing to the exclusive vegetarian menu and the much higher transportation emissions, Astudillo and Azarijafari reported that food and accommodation only accounted for 1% and 2% of the total carbon footprint⁷. Recent studies compared the carbon footprint of in-person and virtual conferences^{4,5,26}, which ranges from 0 to 5.87 kg CO₂ eq. per capita. Among which, Ewijk and Hoekman assumed carbon neutrality for the virtual conference⁴; Jäckle computed the carbon footprint from the electricity needed for devices and servers²⁶; Burtscher et al. considered emissions related to network, laptop, and Zoom-server⁵. Several studies considered multi-site conferences, yet the choices of locations are arbitrary^{4,13}. Stroud and Feeley chose to optimize the conference location by minimizing the carbon footprint while restricting the potential locations to participants' origins⁹. Astudillo and Azarijafari considered the geometric median of all participants as the optimal conference location⁷. As discussed above, none of the existing studies has explicitly explored to what extent virtual conferences and multi-hub hybrid conferences with spatially optimized conference hubs and participant assignments can reduce the environmental impact of in-person conferences.”

References:

- 4 van Ewijk, S. & Hoekman, P. Emission reduction potentials for academic conference travel *Journal of Industrial Ecology* n/a, (2020).
- 5 Burtscher, L. et al. The carbon footprint of large astronomy meetings. *Nature Astronomy* 4, 823-825, (2020).
- 6 Neugebauer, S., Bolz, M., Mankaa, R. & Traverso, M. How sustainable are sustainability conferences? – comprehensive Life Cycle Assessment of an International Conference Series in Europe. *Journal of Cleaner Production* 242, 118516, (2019).
- 7 Astudillo, M. F. & Azarijafari, H. Estimating the global warming emissions of the LCAXVII conference: connecting flights matter. *The International Journal of Life Cycle*

- Assessment 23, 1512-1516, (2018).
- 8 Bossdorf, O., Parepa, M. & Fischer, M. Climate-neutral ecology conferences: just do it! Trends in ecology & evolution 25, 61, (2009).
- 9 Stroud, J. T. & Feeley, K. J. Responsible academia: optimizing conference locations to minimize greenhouse gas emissions. Ecography 38, 402-404, (2015).
- 11 Spinellis, D. & Louridas, P. The Carbon Footprint of Conference Papers. PLOS ONE 8, e66508, (2013).
- 12 Desiere, S. The Carbon Footprint of Academic Conferences: Evidence from the 14th EAAE Congress in Slovenia. EuroChoices 15, 56-61, (2016).
- 13 Hischer, R. & Hilty, L. Environmental impacts of an international conference. Environmental Impact Assessment Review - ENVIRON IMPACT ASSESS REV 22, 543-557, (2002).
- 26 Jäckle, S. Reducing the Carbon Footprint of Academic Conferences by Online Participation: The Case of the 2020 Virtual European Consortium for Political Research General Conference. PS: Political Science & Politics 54, 456-461, (2021).
- 29 Jäckle, S. WE have to change! The carbon footprint of ECPR general conferences and ways to reduce it. European Political Science 18, 630-650, (2019).

On page 2 of the revised Supplementary Information:

“

Supplementary Fig. S1 Comparison of per capita carbon footprint results from previous literature and this study. By taking the whole life cycle of a virtual conference into consideration, the carbon footprint of the virtual conference in this study is substantially higher than those in other studies¹⁻⁴. The carbon footprint of 1-hub in-person conferences is within the range of values reported by existing studies. Only a few studies investigated the carbon footprint of multi-hub in-person conferences. The carbon footprint of 1-hub and 2-hub in-person conferences from Ewijk and Hoekman³ is higher than that from our study because their participants are more distributed than ours. Due to the same reason, the carbon footprint of the 3-hub in-person conference from Ewijk and Hoekman³ is significantly reduced and becomes lower than our result.

Supplementary Table S1 Literature comparison on the life cycle assessment (LCA) of conferences. Abbreviations: GWP, global warming potential; ICT, information and communication technology; CED, cumulative energy demand; LCIA, life cycle impact assessment; NH, New Hampshire; US, the United States; UK, the United Kingdom.

Author, Year	Conference mode	Number of conference hubs	Number of participants	Geographical distribution of participants	Duration (Day)	Location	GWP per capita (kg CO ₂ eq)	Life cycle stages	LCIA method
Jäckle, 2021 ¹	In-person, virtual, hybrid	1	2208	-	5	Innsbruck, Austria	In-person: 566-1166 Virtual: 0.35 (0.03-5.87)	Catering, accommodation, transportation	GWP
Burtscher et al., 2020 ²	In-person, virtual	0,1	1240 (in-person) 1777 (virtual)	84% from Europe (in-person)	5	Lyon, France Virtual	In-person: 1500 Virtual: 0.33	Transportation ICT	GWP
Ewijk and Hoekman, 2020 ³	In-person, virtual, hybrid (1 hub, 10% virtual)	0,1,2,3	625 (in-person) 588 (in-person) 401 (in-person)	Dependent on conference site [‡]	-	Chicago, US Surrey, UK Ulsan, South Korea Virtual	In-person (1 hub): 1513 (1240-1830) In-person (2 hubs): 770 In-person (3 hubs): 450 (330-500) Hybrid: 920-1440 0 (virtual)	Transportation	GWP
Jäckle, 2019 ⁵	In-person, hybrid	1	1930	Europe: 89% North America: 4% Asia: 4% Rest of World: 3%	3-4	Bordeaux, France Glasgow, UK Montreal, Canada Prague, Czechia Oslo, Norway Hamburg, Germany	500-3400	Transportation	GWP
Neugebauer et al., 2019 ⁶	In-person	1	800	Europe: 85% Asia: 5% North America: 4% South America: 3% Rest of World: 3%	3	Aachen, Germany	518-570	Preparation, execution, catering, accommodation, transportation	CML2001, USEtox
Astudillo and Azarijafari, 2018 ⁷	In-person	1	228	North America: 87% Europe: 5% Asia: 4% Rest of World: 4%	3	Portsmouth, NH, US	952	Transportation, catering, accommodation	GWP
Desiere, 2015 ⁸	In-person	1	646	Europe: 85% US: 4% Japan: 2%	-	Ljubljana, Slovenia	150-498	Transportation	GWP

[‡] Most participants are from the continent where the conference is held

				Rest of World: 9%					
Stroud and Feeley, 2014 ⁹	In-person	1	311	-	-	Canary Island, Spain	2580-3220	Transportation	GWP
			207			Merida, Mexico	3000-3540		
			385			Crete, Greece	2510-3020		
			406			Miami, FL, US	2560-2910		
Spinellis and Louridas, 2013 ^{10§}	In-person	-	450	-	-	Bayreuth, Germany	2540-2720	Transportation	GWP
			-			Multi-location multi-year	801 (48 – 2396)		
Bossdorf et al., 2009 ¹¹	In-person	1	125	-	3	Bern, Switzerland	92	Preparation, execution, catering, accommodation, transportation	GWP
Hischier and Hilty, 2002 ⁴	In-person, virtual	0,1,3	308 (in-person) 1000 (virtual)	-	3	Zurich, Switzerland	In-person: 240 Virtual: 2.5 Hybrid: 137	Organization, materials, and transportation	Eco-Indicator 99 UBP 97 CO ₂
This study	In-person, virtual, hybrid	0,1,2,3,4,5,6	536	North America: 88% Europe: 8% Rest of World: 4%		Optimal location	55 (virtual) 455-840 (in-person) 64-1804 (hybrid)	Preparation, execution, catering, accommodation, transportation ICT	GWP CED ReCiPe

”

[§] This paper randomly sampled proceeding conference papers from Scopus

References:

- 1 Jäckle, S. Reducing the Carbon Footprint of Academic Conferences by Online Participation: The Case of the 2020 Virtual European Consortium for Political Research General Conference. *PS: Political Science & Politics* 54, 456-461, (2021).
- 2 Burtscher, L. et al. The carbon footprint of large astronomy meetings. *Nature Astronomy* 4, 823-825, (2020).
- 3 van Ewijk, S. & Hoekman, P. Emission reduction potentials for academic conference travel *Journal of Industrial Ecology* n/a, (2020).
- 4 Hischier, R. & Hilty, L. Environmental impacts of an international conference. *Environmental Impact Assessment Review - ENVIRON IMPACT ASSESS REV* 22, 543-557, (2002).
- 5 Jäckle, S. WE have to change! The carbon footprint of ECPR general conferences and ways to reduce it. *European Political Science* 18, 630-650, (2019).
- 6 Neugebauer, S., Bolz, M., Mankaa, R. & Traverso, M. How sustainable are sustainability conferences? – comprehensive Life Cycle Assessment of an International Conference Series in Europe. *Journal of Cleaner Production* 242, 118516, (2019).
- 7 Astudillo, M. F. & AzariJafari, H. Estimating the global warming emissions of the LCAXVII conference: connecting flights matter. *The International Journal of Life Cycle Assessment* 23, 1512-1516, (2018).
- 8 Desiere, S. The Carbon Footprint of Academic Conferences: Evidence from the 14th EAAE Congress in Slovenia. *EuroChoices* 15, 56-61, (2016).
- 9 Stroud, J. T. & Feeley, K. J. Responsible academia: optimizing conference locations to minimize greenhouse gas emissions. *Ecography* 38, 402-404, (2015).
- 10 Spinellis, D. & Louridas, P. The Carbon Footprint of Conference Papers. *PLOS ONE* 8, e66508, (2013).
- 11 Bossdorf, O., Parepa, M. & Fischer, M. Climate-neutral ecology conferences: just do it! *Trends in ecology & evolution* 25, 61, (2009).

Reviewer's comment (4)

- Concerning the impact from food and heating/electricity at the conference venue vs. participants staying at home. I was wondering whether the estimations assume that at home footprint from food and also from electricity/heating is the same as at the conference venue/conference hotel. In my view this would not make much sense since conference catering or eating at a restaurant probably produces much more food waste, than cooking at home. The same is probably true for heating/electricity: at a large conference venue/hotel I assume the footprint to be higher than at home. I did not find any information on this issue in the paper.

Answer:

Thank you for your comments. The amount of electricity consumption at home and at the hotel is different while food and heating are assumed to be the same. Details were presented in the original manuscript as follows. To address the variation in resource consumption, we have added a sensitivity analysis in the revised manuscript, following the reviewer's suggestions.

On page 21 of the original manuscript:

“The accommodation stage only considers utilities and waste treatment for staying at a hotel or guest home. Specifically, electricity consumption for a guest home is estimated from the U.S. Energy Information Administration's State Energy Data System³⁶, electricity consumption for hotel, thermal energy and water consumption for both guest home and hotel are estimated from previous studies (Supplementary Table S6)^{37,38}.”

References:

- 36 U.S. Energy Information Administration. State Energy Consumption Estimates 1960 Through 2018, Table C17, (June 2020) (Accessed 03/23/2021) <https://www.eia.gov/state/seds/archive/seds2018.pdf>.
- 37 Gössling, S. et al. Tourism and water use: Supply, demand, and security. An international review. *Tourism Management* 33, 1-15, (2012).
- 38 Filimonau, V., Dickinson, J., Robbins, D. & Huijbregts, M. A. J. Reviewing the carbon footprint analysis of hotels: Life Cycle Energy Analysis (LCEA) as a holistic method for carbon impact appraisal of tourist accommodation. *Journal of Cleaner Production* 19, 1917-1930, (2011).

Actions:

On page 19 of the revised manuscript:

“Dietary type as well as the electricity and food consumption rate are the next most sensitive variable for a virtual conference.”

On page 20 of the revised manuscript:

“For virtual conference organizers, advocating energy saving from heating and other residential electricity use (e.g., air conditioning, lighting, electronics, and appliances), food waste reduction, ovo vegetarian diet can be effective practices to improve sustainability.”

On page 21 of the revised Supplementary Information:

“

Supplementary Fig. S18 Sensitivity analysis on food consumption rate and air transportation distance. Virtual and in-person scenarios with 1 to 6 hubs for an average participant are assessed. **a**, Food consumption rate. The base-case food consumption rate is obtained from FAOSTAT^{12,13}, as shown in Table 4. The maximum and minimum total food consumption rate in Table 4 is considered as the best case and worst case. **b**, Air transportation distance. Great-circle distance is considered as the base case, and 110% of the great-circle distance is considered as the worst case²¹.
On page 22 of the revised Supplementary Information:

“

Supplementary Fig. S19 Sensitivity analysis on heat and electricity consumption rate at guest/private home and hotel. Virtual and in-person scenarios with 1 to 6 hubs for an average participant are assessed. **a**, Heat consumption rate. **b**, Electricity consumption rate. The base-, worst-, best-case value of heat and electricity consumption rate at the hotel is obtained from Filimonau et al.²²; The base-, worst-, best-case value of heat and electricity consumption rate at the hotel is obtained from the U.S. Energy Information Administration^{23,24}.

References:

- 12 FAOSTAT. Food Supply - Livestock and Fish Primary Equivalent, (2013) (Accessed 04/19/2021) <http://www.fao.org/faostat/en/#data/CL>.
- 13 FAOSTAT. Food Supply - Crops Primary Equivalent, (2013) (Accessed 04/19/2021) <http://www.fao.org/faostat/en/#data/CC>.

- 21 Kettunen, T. et al. in 6th USA/Europe Air Traffic Management Seminar, Baltimore. 27-30.
- 22 Filimonau, V., Dickinson, J., Robbins, D. & Huijbregts, M. A. J. Reviewing the carbon footprint analysis of hotels: Life Cycle Energy Analysis (LCEA) as a holistic method for carbon impact appraisal of tourist accommodation. *Journal of Cleaner Production* 19, 1917-1930, (2011).
- 23 U.S. Energy Information Administration. State Energy Data System (SEDS): 1960-2019 (complete), (2021) (Accessed 09/11/2021) <https://www.eia.gov/state/seds/seds-data-complete.php?sid=US>.
- 24 U.S. Energy Information Administration. 2015 Residential Energy Consumption Survey (RECS) Data (Accessed 09/11/2021) <https://www.eia.gov/consumption/residential/data/2015/>.

Reviewer's comment (5)

- Concerning the travel induced footprints the authors assume that participants take the train in Europe if the travel distance is below 600 km. I have two questions here: 1) is the travel distance for the train travel calculated by Google Distance Matrix API actually based on the railway network? Or is it based on a street network? Since in many European countries the railway network differs a lot from the street network in terms of density and directness (e.g. in France the high-speed train network connects all major cities with Paris in straight lines, whereas in Germany the train network follows the routes historically from one city to the next, which results often in less direct connections than using the street-network). 2) Assuming a certain threshold for traveling by car/bus/train or by airplane makes sense, yet I doubt that the travel distance itself is the right factor here. I suppose that most conference participants decide primarily on the basis of the travel time.

Thus in cases where a good high-speed train network exists such as in France (e.g. travelling from Marseille to Paris is 775 km by car takes about 8h, by train less than 4h; flying is probably no real option here). Thus, using not a fixed km-threshold, but a threshold for the time accepted for getting to the conference venue might be a better option. See Jäckle (2019, 2021) for a similar approach.

Answer:

Thank you for your comments. For the first question, the travel distance calculated by Google Distance Matrix API is based on the public transit routes if rail is selected as the transit mode (Google Maps Platform, 2021).

Regarding the second question, participants may make decisions on their transportation modes primarily based on the travel time, but this threshold varies based on the traffic, departure time, and route. For example, the bus schedule in Ithaca, New York is different for weekdays and weekends: people need to wait for at most 15 minutes for weekdays and one hour or more to take the bus. And the public transit schedule and traffic conditions (as well as possible flight/train/bus delay) can be very different from location to location. Given the considerable variation in travel time, it can be unfair to set the same departure time or use the default setting of Google Distance Matrix API (i.e., "current" at the time of calling) for the calculation of travel time (Google Maps Platform, 2021). Therefore, we use the travel distance as a threshold in this study to determine the transportation mode of participants. Following the reviewer's suggestions, we have clarified the mechanism of travel distance calculation and the reason for using a distance-based threshold rather than a time-based threshold in the revised manuscript.

References:

Google Maps Platform. Distance Matrix Service, (2021) (Accessed 09/09/2021) <https://developers.google.com/maps/documentation/javascript/distancematrix>.

Actions:

On page 27 of the revised manuscript:

“Notably, the travel distances by rail are computed based on the public transit routes⁶¹. Previous studies made decisions on the transportation modes of participants based on distance-based or time-based thresholds^{6,12,26,29}. Since the travel time computed by Google Distance Matrix API is susceptible to the traffic, departure time, and route, distance-based thresholds are adopted in this study.”

References:

- 6 Neugebauer, S., Bolz, M., Mankaa, R. & Traverso, M. How sustainable are sustainability conferences? – comprehensive Life Cycle Assessment of an International Conference Series in Europe. *Journal of Cleaner Production* 242, 118516, (2019).
- 12 Desiere, S. The Carbon Footprint of Academic Conferences: Evidence from the 14th EAAE Congress in Slovenia. *EuroChoices* 15, 56-61, (2016).
- 26 Jäckle, S. Reducing the Carbon Footprint of Academic Conferences by Online Participation: The Case of the 2020 Virtual European Consortium for Political Research General Conference. *PS: Political Science & Politics* 54, 456-461, (2021).
- 29 Jäckle, S. WE have to change! The carbon footprint of ECPR general conferences and ways to reduce it. *European Political Science* 18, 630-650, (2019).
- 61 Google Maps Platform. Distance Matrix Service, (2021) (Accessed 09/09/2021) <https://developers.google.com/maps/documentation/javascript/distancematrix>.

Reviewer’s comment (6)

- Different organizations report different emission factors for trains, cars and airplanes depending on various assumptions (European Environment Agency, 2021; UK Department for Business, Energy & Industrial Strategy, 2021; Umweltbundesamt, 2021). E.g. on the electricity mix used to power the trains, the average passenger-load factor, whether radiative forcing is included or not for flying etc. This naturally leads to deviating results in the estimation of the overall carbon footprint. The authors did not state – at least I did not find anything on this issue in the paper or the supplement – which assumptions they make, i.e. which emission factors they use for their estimates. Instead of presenting only point estimates it could also make sense to present a minimum-to-maximum range. This might particularly be the case for the travel induced carbon footprint since it clearly – as the results in the paper show – make up by far the highest percentage of emissions for the whole conference attendance. Variations in the emission factors are therefore very relevant for the overall estimates.

Answer:

Thank you for your comments. The characterization factors for air transportation were mentioned in the original Supplementary Information. Specifically, these characterization factors are distance-based (Cox et al., 2018). In terms of rail transportation and driving, characterization factors were extracted from the Ecoinvent database (Tuchs Schmid 2020; ecoinvent 2020). Following the reviewer’s suggestions, all the characterization factors and their corresponding unit processes in the Ecoinvent database have been made available via GitHub (https://github.com/PEESEgroup/Virtual_Con). Moreover, in the revised manuscript, we have

conducted a sensitivity analysis to address the variations in these emission factors, based on the Ecoinvent database and references suggested by the reviewer.

References:

Tuchs Schmid, M., ecoinvent database version 3.7.1. transport, passenger train, DE, Allocation, cut-off by classification, (2020) (Accessed 04/25/2021).

ecoinvent database version 3.7.1. market for transport, passenger car, medium size, petrol, EURO 5, GLO, Allocation, cut-off by classification, (2020) (Accessed 04/25/2021).

On page 9 of the original manuscript:

“The characterization factors for air transportation in Ecoinvent are classified based on four distance categories (i.e., very short-haul, short-haul, medium-haul, and long-haul flights). To be discriminative on the air transportation distances, we adopted the characterization factors from a comprehensive LCA study on air transportation²⁷. Flights with transportation distances of 100 – 1200 km are reported, and characterization factors of air transportation from 1990 to 2050 are assessed and projected by Cox et al.²⁷. We fit the characterization factors of different distances to polynomial functions for the year 2020 and summarized the parameters in Supplementary Table S3.”

References:

27 Cox, B., Jemioło, W. & Mutel, C. Life cycle assessment of air transportation and the Swiss commercial air transport fleet. *Transportation Research Part D: Transport and Environment* 58, 1-13, (2018).

Actions:

On page 19 of the revised manuscript:

“The carbon footprint of in-person scenarios is highly susceptible to the selection of characterization factors for air transportation and the air transportation distances. With more conference hubs, the carbon footprint becomes less variable to changes in these parameters.”

On page 19 of the revised Supplementary Information:

“

Supplementary Fig. S17 Sensitivity analysis for characterization factors of three transportation modes. In-person scenarios with 1 to 6 hubs for an average participant are assessed. The base-case characterization factor for the carbon footprint of air transportation is obtained from Cox et al.¹⁴; the best-case and worst-case characterization factors are obtained from the UK Department for Business, Energy & Industrial Strategy (International flights for an average economy-class passenger and long-haul flights for an average first-class passenger, respectively)¹⁹. The base-, best- and worst-case characterization factors for the carbon footprint of rail transportation and driving are from the Ecoinvent database (Supplementary Table S4) and the Network for Transport Measures^{17,20}. Specifically, the base-, best- and worst-case characterization factors for rail transportation are chosen among the country-specific processes for passenger train or high-speed passenger train, while the selection is conducted among passenger cars with different size and fuel for driving.”

On page 25 of the revised Supplementary Information:

“**Supplementary Table S4** Comparison of characterization factors for the carbon footprint of air transportation, rail transport, and driving. The base-case characterization factor for the carbon footprint of air transportation is obtained from Cox et al.¹⁴; the best-case and worst-case characterization factors are obtained from the UK Department for Business, Energy & Industrial Strategy (International flights for an average economy-class passenger and long-haul flights for an average first-class passenger, respectively)¹⁹. The base-, best- and worst-case characterization

factors for the carbon footprint of rail transportation and driving are from the Ecoinvent database and the Network for Transport Measures^{17,20}. Specifically, the base-, best- and worst-case characterization factors for rail transportation are chosen among the country-specific processes for passenger train or high-speed passenger train, while the selection is conducted among passenger cars with different sizes and fuel for driving.

Category	Item	Location	Value	Unit	Case
Air transportation	Distance-based ³ , see Supplementary Table S3	Swiss	-	-	Base
	International flights, economy class ¹⁹	UK	1.39E-01	kg CO ₂ eq./pkm	Best
	Long-haul flights, first class ¹⁹	UK	5.85E-01	kg CO ₂ eq./pkm	Worst
	transport, passenger train, DE ¹⁷	Germany	7.45E-02	kg CO ₂ eq./pkm	Base
	transport, passenger train, BE ¹⁷	Belgium	4.53E-02	kg CO ₂ eq./pkm	
	transport, passenger train, FR ¹⁷	France	2.37E-02	kg CO ₂ eq./pkm	
	transport, passenger train, IT ¹⁷	Italy	4.89E-02	kg CO ₂ eq./pkm	
	transport, passenger train, RoW ¹⁷	Rest-of-World	7.66E-02	kg CO ₂ eq./pkm	Worst
	transport, passenger train, high-speed, DE ¹⁷	Germany	6.10E-02	kg CO ₂ eq./pkm	
	transport, passenger train, high-speed, FR ¹⁷	France	2.00E-02	kg CO ₂ eq./pkm	
Rail transport	transport, passenger train, high-speed, IT ¹⁷	Italy	4.88E-02	kg CO ₂ eq./pkm	
	transport, passenger train, high-speed, RoW ¹⁷	Rest-of-World	7.14E-02	kg CO ₂ eq./pkm	
	High speed train with green electricity ²⁰	-	5.70E-04	kg CO ₂ eq./pkm	Best
	High speed train ²⁰	EU	3.76E-02	kg CO ₂ eq./pkm	
	Inter city train with green electricity ²⁰	-	7.10E-04	kg CO ₂ eq./pkm	
	Inter city train ²⁰	EU	4.63E-02	kg CO ₂ eq./pkm	
	Regional train with green electricity ²⁰	-	8.00E-04	kg CO ₂ eq./pkm	
	Regional train ²⁰	EU	5.25E-02	kg CO ₂ eq./pkm	
	market for transport, passenger car, small size, petrol, EURO 5, GLO ¹⁷	Global	2.71E-01	kg CO ₂ eq./pkm	
	market for transport, passenger car, small size, diesel, EURO 5, GLO ¹⁷	Global	2.34E-01	kg CO ₂ eq./pkm	
Driving	market for transport, passenger car, medium size, petrol, EURO 5, GLO ¹⁷	Global	3.40E-01	kg CO ₂ eq./pkm	Base
	market for transport, passenger car, medium size, diesel, EURO 5, GLO ¹⁷	Global	3.09E-01	kg CO ₂ eq./pkm	
	market for transport, passenger car, large size, petrol, EURO 5, GLO ¹⁷	Global	4.09E-01	kg CO ₂ eq./pkm	Worst
	market for transport, passenger car, large size, diesel, EURO 5, GLO ¹⁷	Global	3.86E-01	kg CO ₂ eq./pkm	
	market for transport, passenger car, electric, GLO ¹⁷	Global	2.26E-01	kg CO ₂ eq./pkm	Best

”

On page 30 of the revised manuscript:

“All data needed to evaluate the conclusions in the paper are present in the paper or can be accessed through the Supplementary Materials and GitHub (https://github.com/PEESEgroup/Virtual_Con) with the published version archived (<https://doi.org/10.5281/zenodo.5515049>).”

References:

14 Cox, B., Jemiolo, W. & Mutel, C. Life cycle assessment of air transportation and the Swiss commercial air transport fleet. *Transportation Research Part D: Transport and Environment* 58, 1-13, (2018).

17 Wernet, G. et al. The ecoinvent database version 3 (part I): overview and methodology. *Int. J. Life Cycle Assess.* 21, 1218-1230, (2016).

19 UK Department for Business, Energy & Industrial Strategy. *Greenhouse Gas Reporting: Conversion Factors 2020*, (2020) (Accessed 09/11/2021) <https://www.gov.uk/government/publications/greenhouse-gas-reporting-conversion-factors-2020>.

20 Network for Transport Measures. *Default and Benchmark Transport Data.*, (2018) (Accessed 09/11/2021) <https://www.transportmeasures.org/en/wiki/evaluation-transport-suppliers/train-travel-baselines-2018/>.

Reviewer’s comment (7)

- Since air-travel is by far the biggest component of the carbon footprint of conferences, it makes sense to be as exact as possible here. I was wondering whether the distances used to calculate the carbon emissions from flying are solely the great circle distances between the airports or whether

the fact that many airplanes do not fly the shortest route but more inefficient detours, or have stop overs has been included in the estimations. It has been shown that the actual distances aircraft fly are between 6 and 10% longer than the great circle routes between the departure and the destination airports (Kettunen et al., 2005). Furthermore, is there a difference in the estimation between long-haul and short-haul flights? Since long-haul flights reach higher altitudes where CO2 exerts more harmful effects it may makes sense to make a distinction. Does the analysis account for the different electricity mix in the participants' countries for those joining virtually?

Answer:

Thank you for your comments. We calculate the air transportation distances solely based on the great circle distances without considering the more inefficient detours or stopovers. Following the reviewer's suggestions, we have added a sensitivity analysis on the air transportation distances. In terms of estimating for long-haul and short-haul flights, we adopted the distance-based characterization factors from an existing study (Cox et al., 2018). In particular, the value of the characterization factor decreases as the distance increases, which can be fitted by power functions shown in Supplementary Fig. S2 and Supplementary Table S2 of the original manuscript. This study considered the impacts of greenhouse gas emitted in the upper troposphere and lower stratosphere. We also account for the local power grids of each virtual participant.

References:

Cox, B., Jemiolo, W. & Mutel, C. Life cycle assessment of air transportation and the Swiss commercial air transport fleet. Transportation Research Part D: Transport and Environment 58, 1-13, (2018).

On page 4 of the original Supplementary Information:

“

Supplementary Fig. S2 The carbon footprint and CED per passenger-km (pkm) of flight. We fit the characterization factors of air transportation over distance from Cox et al.³ to polynomial functions for the year 2020. The values of power, slope, intercept, and R^2 are summarized in Supplementary Table S2.

Supplementary Table S2 Parameters of the fitted power functions for the characterization factors of air transportation over distance from Cox et al.³.

Impact category	Power	Slope	Intercept	R^2
Global Warming Potential (kg CO ₂ eq per passenger km)	-8.39E-01	4.06E+01	1.12E-01	9.97E-01
Cumulative Energy Demand, Non-Renewable (MJ per passenger km)	-8.82E-01	6.08E+02	1.05E+00	9.98E-01
Agricultural Land Occupation Potential (m ² yr per passenger km)	-8.94E-01	4.29E-01	2.52E-04	9.99E-01
Fossil Depletion Potential (kg oil eq per passenger km)	-8.80E-01	1.39E+01	2.48E-02	9.98E-01
Freshwater Ecotoxicity Potential (kg 14-DCB eq per passenger km)	-9.19E-01	1.08E-01	8.57E-05	9.99E-01
Freshwater Eutrophication Potential (kg P eq per passenger km)	-9.57E-01	3.43E-03	1.30E-06	1.00E+00
Human Toxicity Potential (kg 1,4 DB eq per passenger km)	-9.83E-01	5.92E+00	2.07E-03	1.00E+00
Ionising Radiation Potential (kg U235 eq per passenger km)	-9.02E-01	3.67E+00	4.79E-03	9.99E-01
Marine Ecotoxicity Potential (kg 14-DCB eq per passenger km)	-9.30E-01	1.06E-01	6.63E-05	9.99E-01
Marine Eutrophication Potential (kg N eq per passenger km)	-9.20E-01	6.03E-02	1.38E-04	1.00E+00
Metal Depletion Potential (kg Fe eq per passenger km)	-9.15E-01	3.58E-01	3.45E-04	9.99E-01
Natural Land Transformation Potential (m ² per passenger km)	-8.81E-01	1.48E-02	2.60E-05	9.98E-01
Ozone Depletion Potential (kg CFC11 eq per passenger km)	-8.78E-01	7.00E-06	1.26E-08	9.98E-01
Particulate Matter Formation Potential (kg PM10 eq per passenger km)	-9.36E-01	4.31E-02	2.75E-05	1.00E+00
Photochemical Oxidant Formation Potential (kg NMVOC per passenger km)	-9.58E-01	1.65E-01	6.36E-05	1.00E+00
Terrestrial Acidification Potential (kg SO ₂ eq per passenger km)	-9.57E-01	2.32E-03	1.15E-06	1.00E+00

Terrestrial Ecotoxicity Potential (kg 14-DCB eq per passenger km)	-9.57E-01	2.32E-03	1.15E-06	1.00E+00
Urban Land Occupation Potential (m ² yr per passenger km)	-9.66E-01	3.78E-01	1.30E-04	1.00E+00
Water Depletion Potential (m ³ H ₂ O per passenger km)	-9.43E-01	1.17E-01	7.29E-05	1.00E+00

”

Actions:

On page 19 of the revised manuscript:

“The carbon footprint of in-person scenarios is highly susceptible to the selection of characterization factors for air transportation and the air transportation distances. With more conference hubs, the carbon footprint becomes less variable to changes in these parameters. This result suggests that participants should make efforts to take flights with as few stopovers as possible.”

On page 21 of the revised Supplementary Information:

“

Supplementary Fig. S18 Sensitivity analysis on food consumption rate and air transportation distance. Virtual and in-person scenarios with 1 to 6 hubs for an average participant are assessed. **a**, Food consumption rate. The base-case food consumption rate is obtained from FAOSTAT^{12,13}, as shown in Table 4. The maximum and minimum total food consumption rate in Table 4 is considered as the best case and worst case. **b**, Air transportation distance. Great-circle distance is considered as the base case, and 110% of the great-circle distance is considered as the worst case²¹.”

References:

- 12 FAOSTAT. Food Supply - Livestock and Fish Primary Equivalent, (2013) (Accessed 04/19/2021) <http://www.fao.org/faostat/en/#data/CL>.
- 13 FAOSTAT. Food Supply - Crops Primary Equivalent, (2013) (Accessed 04/19/2021) <http://www.fao.org/faostat/en/#data/CC>.
- 21 Kettunen, T. et al. in 6th USA/Europe Air Traffic Management Seminar, Baltimore. 27-30.

Reviewer’s comment (8)

• *Fig 2c is difficult to read. The colors/shading of the legend should be enlarged.*

Answer:

Thank you for your comments. We have revised Fig. 2c in the revised manuscript, following the reviewer’s suggestions.

Actions:

On page 9 of the revised manuscript:

“

Fig. 2 Comparison between the 1-hub in-person conference and the virtual conference. a, Carbon footprint associated with transportation for individual participants, indicating that the per pkm carbon footprint for trips primarily by plane (slope of cyan dashed line) tends to be smaller than that for driving (slope of orange dashed line). **b,** Cumulative carbon footprint for participants

with increasing travel distances, showing that 50% of the GHG emissions for all participants' trips are resulted from long-distance travels with a round-trip distance of over 10,000 km. c, Environmental profiles of the virtual conference and in-person conference with only one hub, suggesting the environmental and energy sustainability of virtual conference in all impact categories. Specifically, transforming in-person conferences to pure virtual mode dramatically reduces the carbon footprint by 94% and CED by 90%. Among the impact categories at the midpoint level, air transportation dominates fossil depletion, marine eutrophication, natural land transformation, ozone depletion, and photochemical oxidant formation for the 1-hub in-person scenario. And food preparation, electricity consumption at home, and ICT services contribute to the majority of each impact category for the baseline virtual scenario. Moreover, agricultural land occupation, terrestrial ecotoxicity, and water depletion are dominated by food preparation for both in-person and virtual conferences.”

Trend of virtual and hybrid conferences since COVID-19 effectively mitigate climate change

Yanqiu Tao¹, Debbie Steckel², Fengqi You^{1,3*}

¹ Systems Engineering, Cornell University, Ithaca, New York, 14853, USA

² The American Center for Life Cycle Assessment, Bethesda, Maryland, 20824, USA

³ Robert Frederick Smith School of Chemical and Biomolecular Engineering, Cornell University, Ithaca, NY, 14853, USA

Response to reviewers and actions taken

Reviewer 3

The authors are most grateful to the editor for the helpful comments.

Reviewer's comment (1)

Introduction, I would extend the introduction or added a paragraph on the current state of the art on LCA and Carbon Footprint of conferences, to clearly highlight the innovative aspects of the current studies. You cited some studies with 4-13 references, I would add at least a table and a chapter indicating the main content in the previous papers.

Answer:

Thank you for your comments. In the revised manuscript, we have added a Comparison with the state-of-the-art conference LCA studies subsection to summarize the results and assumptions of existing studies. We have also added a table in the Supplementary Information, following the reviewer's suggestions.

Actions:

On page 6 of the revised manuscript:

“Comparison with the state-of-the-art conference LCA studies

Supplementary Table S1 summarizes and compares the state-of-the-art conference LCA studies. Previous LCA studies mainly focused on quantifying the carbon footprint of in-person conferences, while only two of them presented results for other impact categories using the life cycle impact assessment (LCIA) method, including CML2001, USEtox, Eco-Indicator 99, and UBP 97^{6,13}. Half of them focused exclusively on round-trip transportation^{4,5,9,11,12,29} while the other half considered life cycle stages of preparation, execution, catering, accommodation, and transportation^{6-8,13,26}. However, due to differences in assumptions associated with in-person conferences (e.g., duration, size, and locations of the conference, geographical distribution of participants, transportation mode, system boundary, and selection of characterization factors), the carbon footprint ranges from 92 to 3540 kg CO₂ eq. per capita. All of these studies identified transportation as the environmental hotspot. The conference site and geographical distribution of participants determine the transportation distance and mode for participants. From those who reported the average transportation distance, the average round-trip transportation distance varies from 1980 km¹³ to

9564 km⁹. However, the average distance does not illustrate the complete picture of the participant transportation, and it was found that the 10–20% of participants with the most polluting trips contribute to a substantial portion (20–70%) of the total transportation-induced emissions^{4-7,12}. These values depend on the distribution of participants, which reveals whether the conference is more localized or more internationalized. As shown in Supplementary Table S1, most participants are from the region where the conferences are held. The conference location is also important in determining the transportation profile, in which a conference location with better train connection to other major cities is capable of allowing more participants to transport by train and thus has more potential of reducing carbon footprint while a conference located in the southern hemisphere usually perform much worse in terms of carbon footprint compared to the northern hemisphere^{11,12}. Bossdorf et al. suggested food and accommodation accounted for 18% and 13% of the total carbon footprint, respectively⁸. On the other hand, owing to the exclusive vegetarian menu and the much higher transportation emissions, Astudillo and Azarijafari reported that food and accommodation only accounted for 1% and 2% of the total carbon footprint⁷. Recent studies compared the carbon footprint of in-person and virtual conferences^{4,5,26}, which ranges from 0 to 5.87 kg CO₂ eq. per capita. Among which, Ewijk and Hoekman assumed carbon neutrality for the virtual conference⁴; Jäckle computed the carbon footprint from the electricity needed for devices and servers²⁶; Burtscher et al. considered emissions related to network, laptop, and Zoom-server⁵. Several studies considered multi-site conferences, yet the choices of locations are arbitrary^{4,13}. Stroud and Feeley chose to optimize the conference location by minimizing the carbon footprint while restricting the potential locations to participants' origins⁹. Astudillo and Azarijafari considered the geometric median of all participants as the optimal conference location⁷. As discussed above, none of the existing studies has explicitly explored to what extent virtual conferences and multi-hub hybrid conferences with spatially optimized conference hubs and participant assignments can reduce the environmental impact of in-person conferences.”

References:

- 4 van Ewijk, S. & Hoekman, P. Emission reduction potentials for academic conference travel *Journal of Industrial Ecology* n/a, (2020).
- 5 Burtscher, L. et al. The carbon footprint of large astronomy meetings. *Nature Astronomy* 4, 823-825, (2020).
- 6 Neugebauer, S., Bolz, M., Mankaa, R. & Traverso, M. How sustainable are sustainability conferences? – comprehensive Life Cycle Assessment of an International Conference Series in Europe. *Journal of Cleaner Production* 242, 118516, (2019).
- 7 Astudillo, M. F. & Azarijafari, H. Estimating the global warming emissions of the LCAXVII conference: connecting flights matter. *The International Journal of Life Cycle Assessment* 23, 1512-1516, (2018).
- 8 Bossdorf, O., Parepa, M. & Fischer, M. Climate-neutral ecology conferences: just do it! *Trends in ecology & evolution* 25, 61, (2009).
- 9 Stroud, J. T. & Feeley, K. J. Responsible academia: optimizing conference locations to minimize greenhouse gas emissions. *Ecography* 38, 402-404, (2015).
- 11 Spinellis, D. & Louridas, P. The Carbon Footprint of Conference Papers. *PLOS ONE* 8, e66508, (2013).
- 12 Desiere, S. The Carbon Footprint of Academic Conferences: Evidence from the 14th EAAE Congress in Slovenia. *EuroChoices* 15, 56-61, (2016).
- 13 Hischer, R. & Hilty, L. Environmental impacts of an international conference. *Environmental Impact Assessment Review - ENVIRON IMPACT ASSESS REV* 22, 543-

- 557, (2002).
- 26 Jäckle, S. Reducing the Carbon Footprint of Academic Conferences by Online Participation: The Case of the 2020 Virtual European Consortium for Political Research General Conference. *PS: Political Science & Politics* 54, 456-461, (2021).
- 29 Jäckle, S. WE have to change! The carbon footprint of ECPR general conferences and ways to reduce it. *European Political Science* 18, 630-650, (2019).

On page 2 of the revised Supplementary Information:

“

Supplementary Fig. S1 Comparison of per capita carbon footprint results from previous literature and this study. By taking the whole life cycle of a virtual conference into consideration, the carbon footprint of the virtual conference in this study is substantially higher than those in other studies¹⁻⁴. The carbon footprint of 1-hub in-person conferences is within the range of values reported by existing studies. Only a few studies investigated the carbon footprint of multi-hub in-person conferences. The carbon footprint of 1-hub and 2-hub in-person conferences from Ewijk and Hoekman³ is higher than that from our study because their participants are more distributed than ours. Due to the same reason, the carbon footprint of the 3-hub in-person conference from Ewijk and Hoekman³ is significantly reduced and becomes lower than our result.

Supplementary Table S1 Literature comparison on the life cycle assessment (LCA) of conferences. Abbreviations: GWP, global warming potential; ICT, information and communication technology; CED, cumulative energy demand; LCIA, life cycle impact assessment; NH, New Hampshire; US, the United States; UK, the United Kingdom.

Author, Year	Conference mode	Number of conference hubs	Number of participants	Geographical distribution of participants	Duration (Day)	Location	GWP per capita (kg CO ₂ eq)	Life cycle stages	LCIA method
Jäckle, 2021 ¹	In-person, virtual, hybrid	1	2208	-	5	Innsbruck, Austria	In-person: 566-1166 Virtual: 0.35 (0.03-5.87)	Catering, accommodation, transportation	GWP
Burtscher et al., 2020 ²	In-person, virtual	0,1	1240 (in-person) 1777 (virtual)	84% from Europe (in-person)	5	Lyon, France Virtual	In-person: 1500 Virtual: 0.33	Transportation ICT	GWP
Ewijk and Hoekman, 2020 ³	In-person, virtual, hybrid (1 hub, 10% virtual)	0,1,2,3	625 (in-person) 588 (in-person) 401 (in-person)	Dependent on conference site**	-	Chicago, US Surrey, UK Ulsan, South Korea Virtual	In-person (1 hub): 1513 (1240-1830) In-person (2 hubs): 770 In-person (3 hubs): 450 (330-500) Hybrid: 920-1440 0 (virtual)	Transportation	GWP
Jäckle, 2019 ⁵	In-person, hybrid	1	1930	Europe: 89% North America: 4% Asia: 4% Rest of World: 3%	3-4	Bordeaux, France Glasgow, UK Montreal, Canada Prague, Czechia Oslo, Norway Hamburg, Germany	500-3400	Transportation	GWP
Neugebauer et al., 2019 ⁶	In-person	1	800	Europe: 85% Asia: 5% North America: 4% South America: 3% Rest of World: 3%	3	Aachen, Germany	518-570	Preparation, execution, catering, accommodation, transportation	CML2001, USEtox
Astudillo and Azarjafari, 2018 ⁷	In-person	1	228	North America: 87% Europe: 5% Asia: 4% Rest of World: 4%	3	Portsmouth, NH, US	952	Transportation, catering, accommodation	GWP
Desiere, 2015 ⁸	In-person	1	646	Europe: 85% US: 4% Japan: 2%	-	Ljubljana, Slovenia	150-498	Transportation	GWP

** Most participants are from the continent where the conference is held

				Rest of World: 9%					
Stroud and Feeley, 2014 ⁹	In-person	1	311	-	-	Canary Island, Spain	2580-3220	Transportation	GWP
			207			Merida, Mexico	3000-3540		
			385			Crete, Greece	2510-3020		
			406			Miami, FL, US	2560-2910		
Spinellis and Louridas, 2013 ^{10††}	In-person	-	450	-	-	Bayreuth, Germany	2540-2720	Transportation	GWP
			-			Multi-location multi-year	801 (48 – 2396)		
Bossdorf et al., 2009 ¹¹	In-person	1	125	-	3	Bern, Switzerland	92	Preparation, execution, catering, accommodation, transportation	GWP
Hischier and Hilty, 2002 ⁴	In-person, virtual	0,1,3	308 (in-person) 1000 (virtual)	-	3	Zurich, Switzerland	In-person: 240 Virtual: 2.5 Hybrid: 137	Organization, materials, and transportation	Eco-Indicator 99 UBP 97 CO ₂
This study	In-person, virtual, hybrid	0,1,2,3,4,5,6	536	North America: 88% Europe: 8% Rest of World: 4%		Optimal location	55 (virtual) 455-840 (in-person) 64-1804 (hybrid)	Preparation, execution, catering, accommodation, transportation ICT	GWP CED ReCiPe

”

†† This paper randomly sampled proceeding conference papers from Scopus

References:

- 1 Jäckle, S. Reducing the Carbon Footprint of Academic Conferences by Online Participation: The Case of the 2020 Virtual European Consortium for Political Research General Conference. *PS: Political Science & Politics* 54, 456-461, (2021).
- 2 Burtscher, L. et al. The carbon footprint of large astronomy meetings. *Nature Astronomy* 4, 823-825, (2020).
- 3 van Ewijk, S. & Hoekman, P. Emission reduction potentials for academic conference travel *Journal of Industrial Ecology* n/a, (2020).
- 4 Hischier, R. & Hilty, L. Environmental impacts of an international conference. *Environmental Impact Assessment Review - ENVIRON IMPACT ASSESS REV* 22, 543-557, (2002).
- 5 Jäckle, S. WE have to change! The carbon footprint of ECPR general conferences and ways to reduce it. *European Political Science* 18, 630-650, (2019).
- 6 Neugebauer, S., Bolz, M., Mankaa, R. & Traverso, M. How sustainable are sustainability conferences? – comprehensive Life Cycle Assessment of an International Conference Series in Europe. *Journal of Cleaner Production* 242, 118516, (2019).
- 7 Astudillo, M. F. & AzariJafari, H. Estimating the global warming emissions of the LCAXVII conference: connecting flights matter. *The International Journal of Life Cycle Assessment* 23, 1512-1516, (2018).
- 8 Desiere, S. The Carbon Footprint of Academic Conferences: Evidence from the 14th EAAE Congress in Slovenia. *EuroChoices* 15, 56-61, (2016).
- 9 Stroud, J. T. & Feeley, K. J. Responsible academia: optimizing conference locations to minimize greenhouse gas emissions. *Ecography* 38, 402-404, (2015).
- 10 Spinellis, D. & Louridas, P. The Carbon Footprint of Conference Papers. *PLOS ONE* 8, e66508, (2013).
- 11 Bossdorf, O., Parepa, M. & Fischer, M. Climate-neutral ecology conferences: just do it! *Trends in ecology & evolution* 25, 61, (2009).

Reviewer’s comment (2)

The paper has lots of information, but it is not always clear reading it which are the data you have used and there are some Life cycle inventory data missing in the main manuscript. I know that the most are reported in the supplementary materials but it doesn’t help the transparency of the study. Please report the main data of life cycle inventory in the main manuscript: e.g. the consumption of each transport system considered, the FAO data on Food, and so on.

Answer:

Thank you for your comments. Following the reviewer’s suggestions, we have moved the main life cycle inventory data from the original Supplementary Information to the Methods section of the revised manuscript.

Actions:

On page 22 of the revised manuscript:

“Table 1 Energy and water consumption of food preparation per capita per day⁶.

Item	Value	Unit
Electricity	4.00	kWh
Heat	6.00	MJ
Water	28.00	kg
Wastewater	0.028	m ³

Table 2 Energy and water consumption of accommodation per capita per day by the type of accommodation^{45-47,50,51}.

Type of accommodation	Item	Value	Unit
Guest home	Electricity	12.02	kWh
	Heat	49.25	MJ
	Water	651.85	kg
	Wastewater	0.65	m ³
Hotel	Electricity	18.10	kWh
	Heat	18.36	MJ
	Water	900.00	kg
	Wastewater	0.90	m ³

Table 3 Average amount, composition, and management pathways of municipal solid waste (MSW) in the U.S. per capita per day⁴⁸.

Category	2018 MSW generation (kg)	Weight recycled (kg)	Weight composted (kg)	Weight other food management pathways (kg)	Weight combusted with energy recovery (kg)	Weight landfilled (kg)
Paper and paperboard	0.56	0.38	-	-	0.04	0.14
Glass	0.10	0.03	-	-	0.01	0.06
Steel	0.16	0.05	-	-	0.02	0.09
Aluminium, other nonferrous metals	0.05	0.02	-	-	0.01	0.03
Plastics	0.30	0.03	-	-	0.05	0.23
Rubber and leather	0.08	0.01	-	-	0.02	0.04
Textiles	0.14	0.02	-	-	0.03	0.09
Wood	0.15	0.03	-	-	0.02	0.10
Food, yard trimmings, other MSW organics	0.83	-	0.21	0.15	0.08	0.38
Other MSW	0.07	0.01	-	-	0.01	0.05
Total	2.45	0.58	0.21	0.15	0.29	1.22

”

On page 24 of the revised manuscript:

“Table 5 Material and energy consumption of preparation and execution^{6,13,52}.

Category	Item	Specification	Value	Unit
Preparation - local conference committee	Computer usage	operation, computer, laptop, 68% active work	2.00E-01	h
		operation, computer, laptop, 23% active work	2.00E-01	h
		operation, computer, laptop, standby/sleep mode	2.00E-01	h
	Printed matter, abstract reviewing process	Graphic paper, 100% recycled	7.04E-03	kg
		Operation of printer, laser, black and white, per kg printed matter	7.04E-03	kg
	Printed matter, meeting	Graphic paper, 100% recycled	3.00E-03	kg
		Operation of printer, laser, black and white, per kg printed matter	3.00E-03	kg
	Waste paper	Paper waste disposal	1.00E-02	kg
Preparation - permanent conference committee	Computer usage	operation, computer, laptop, 68% active work	1.40E-01	h
		operation, computer, laptop, 23% active work	1.40E-01	h
		operation, computer, laptop, standby/sleep mode	1.40E-01	h
	Operation of printer, laser, black and white, per kg printed matter	Graphic paper, 100% recycled	1.31E-03	kg
		Operation of printer, laser, black and white, per kg printed matter	1.31E-03	kg
	Waste paper	Paper waste disposal	1.31E-03	kg
Preparation - Participants	Abstract writing	Operation, computer, laptop, active mode	7.05E-01	h
	Poster printing	Paper	4.65E-03	kg
		Operation, printer, laser, colour, per kg printed paper	4.65E-03	kg
	Slide preparation	Operation, computer, laptop, active mode	5.35E-01	h
Preparation - Others	Office work - secretariat	Operation, computer, laptop, active mode	2.70E+00	h
	Website maintenance	Operation, computer, laptop, active mode	1.62E+00	h
	Conference program	Paper, wood containing, supercalendered	3.86E-01	kg
		operation, printer, laser, colour, per kg printed paper	3.86E-01	kg
	Proceedings booklet	Paper, wood containing, supercalendered	3.41E+00	kg
		operation, printer, laser, colour, per kg printed paper	3.41E+00	kg
		Jute textile production	7.97E-02	kg
Jute bag	Transport, cargo aircraft, intercontinental		5.90E-01	ton-km
		Transport, freight, truck	2.39E-02	ton-km
Execution - Conference meeting rooms	Electricity	Electricity production	2.84E+01	kWh
	Water	Tap water production	8.68E+01	kg
	Wastewater	Wastewater treatment	8.68E-02	m ³
	Waste	Waste treatment	1.22E+00	kg

”

On page 26 of the revised manuscript:

“Table 6 Material and energy consumption of the information and communication technology for virtual-conferencing^{5,39,43,53,54,56,57}.

Category	Item	Value	Unit
Device	Router, internet	1.09E-01	unit
	Internet access equipment	1.03E+00	unit
	Computer, laptop	1.54E+00	unit
Device-related electricity	Energy consumption for router	0.00E+00	kWh
	Energy consumption for internet access equipment (ADSL, DSLAM)	0.00E+00	kWh
	Energy consumption for laptop, video mode	3.43E+02	kWh
Network-related electricity	PS-Core	2.38E+00	kWh
	Fixed line CPE	3.83E+00	kWh
	Operator DC	4.75E-01	kWh
	Office networks	1.18E+00	kWh
	Internet core	2.88E-01	kWh
	Infrastructure	6.67E-01	kWh
Server-related electricity	Network	5.56E-02	kWh
	Storage	2.22E-01	kWh
	Server	1.07E+00	kWh

References:

- 5 Burtscher, L. et al. The carbon footprint of large astronomy meetings. *Nature Astronomy* 4, 823-825, (2020).
- 6 Neugebauer, S., Bolz, M., Mankaa, R. & Traverso, M. How sustainable are sustainability

- conferences? – comprehensive Life Cycle Assessment of an International Conference Series in Europe. *Journal of Cleaner Production* 242, 118516, (2019).
- 13 Hischier, R. & Hilty, L. Environmental impacts of an international conference. *Environmental Impact Assessment Review - ENVIRON IMPACT ASSESS REV* 22, 543-557, (2002).
- 30 FAOSTAT. Food Supply - Crops Primary Equivalent, (2013) (Accessed 04/19/2021) <http://www.fao.org/faostat/en/#data/CC>.
- 31 FAOSTAT. Food Supply - Livestock and Fish Primary Equivalent, (2013) (Accessed 04/19/2021) <http://www.fao.org/faostat/en/#data/CL>.
- 39 Wernet, G. et al. The ecoinvent database version 3 (part I): overview and methodology. *Int. J. Life Cycle Assess.* 21, 1218-1230, (2016).
- 43 Andrae, A. S. G. & Edler, T. On Global Electricity Usage of Communication Technology: Trends to 2030. *Challenges* 6, (2015).
- 45 U.S. Energy Information Administration. State Energy Data System (SEDS): 1960-2019 (complete), (2021) (Accessed 09/11/2021) <https://www.eia.gov/state/seds/seds-data-complete.php?sid=US>.
- 46 Gössling, S. et al. Tourism and water use: Supply, demand, and security. An international review. *Tourism Management* 33, 1-15, (2012).
- 47 Filimonau, V., Dickinson, J., Robbins, D. & Huijbregts, M. A. J. Reviewing the carbon footprint analysis of hotels: Life Cycle Energy Analysis (LCEA) as a holistic method for carbon impact appraisal of tourist accommodation. *Journal of Cleaner Production* 19, 1917-1930, (2011).
- 48 United States Environmental Protection Agency. Advancing Sustainable Materials Management: 2018 Fact Sheet, Table 1, (December 2020) (Accessed 03/23/2021) https://www.epa.gov/sites/production/files/2020-11/documents/2018_ff_fact_sheet.pdf.
- 50 U.S. Energy Information Administration. 2015 Residential Energy Consumption Survey (RECS) Data (Accessed 09/11/2021) <https://www.eia.gov/consumption/residential/data/2015/>.
- 51 Cheryl, A. D. et al., U.S. Geological Survey. Estimated Use of Water in the United States in 2015, (2018) (Accessed 04/25/2021) <https://pubs.usgs.gov/circ/1441/circ1441.pdf>.
- 52 GREENVIEW. 2017 Green Venue Report: The State of Convention & Exhibition Center Sustainability, (2017) (Accessed 03/23/2021) <https://greenview.sg/wp-content/uploads/2017/08/green-venue-report-2017.pdf>.
- 53 Pärssinen, M., Kotila, M., Cuevas, R., Phansalkar, A. & Manner, J. Environmental impact assessment of online advertising. *Environmental Impact Assessment Review* 73, 177-200, (2018).
- 54 Shehabi, A., Smith, S. J., Masanet, E. & Koomey, J. Data center growth in the United States: decoupling the demand for services from electricity use. *Environmental Research Letters* 13, 124030, (2018).
- 56 Malmodin, J. & Lundén, D. The energy and carbon footprint of the ICT and E&M sector in Sweden 1990-2015 and beyond. (2016).
- 57 Zoom Help Center. System requirements for Windows, macOS, and Linux (Accessed 03/23/2021) <https://support.zoom.us/hc/en-us/articles/201362023>.

Reviewer's comment (3)

The sequence of the manuscript is also not conformed to the an LCA study in according to the ISO 14040: Methodology with all 4 phases of LCA should be reported after the introduction and state of the art to allow the reader to understand the environmental impacts obtained. It is very important for an LCA study to be transparent in its assumptions made and ensuring transparency and reproducibility.

Answer:

Thank you for your comments. The sequence of the manuscript is organized according to Nature Communication's guide for submission (<https://www.nature.com/ncomms/submit/article>), where the Methods section appears after the Results and Discussion sections of the main text. Following the reviewer's suggestions, we have added a sentence to the Introduction section of the revised manuscript to direct the readers to the Methods section for the details of the four phases of life cycle assessment (LCA).

Actions:

On page 4 of the revised manuscript:

“Following the ISO 14040 standard²⁸, the LCA methodology is constituted by four phases, including goal and scope definition, life cycle inventory, life cycle impact assessment, and interpretation. The four phases of this study are detailed in the Methods section.”

References:

- 28 International Standards Organization. ISO 14040:2006 Environmental management — Life cycle assessment — Principles and framework, (2006).

Reviewer's comment (4)

In page 7 Fig. 2 Comparison between the 1-hub in-person conference and the virtual conference. a, Carbon footprint associated with transportation for individual participants, indicating that the unit carbon footprint for trips primarily by plane tends to be smaller than that for driving. I do not see this result in the figure and it is not coherent with the results of other studies, please clarify it.

Answer:

Thank you for your comments. Unit carbon footprint means carbon footprint per passenger-kilometer (pkm) for each participant. In Fig. 2a, the slope of the fitted line for participants traveled by car (red color) is larger than the slope of the fitted line for participants traveled primarily by planes (green color). Therefore, the unit carbon footprint for trips primarily by plane tends to be smaller than that for driving. We have revised Fig 2a and its legend in the revised manuscript to avoid confusion, following the reviewer's suggestions.

Actions:

On page 9 of the revised manuscript:

“

Fig. 2 Comparison between the 1-hub in-person conference and the virtual conference. a, Carbon footprint associated with transportation for individual participants, indicating that the per pkm carbon footprint for trips primarily by plane (slope of cyan dashed line) tends to be smaller than that for driving (slope of orange dashed line). **b,** Cumulative carbon footprint for participants

with increasing travel distances, showing that 50% of the GHG emissions for all participants' trips are resulted from long-distance travels with a round-trip distance of over 10,000 km. c, Environmental profiles of the virtual conference and in-person conference with only one hub, suggesting the environmental and energy sustainability of virtual conference in all impact categories. Specifically, transforming in-person conferences to pure virtual mode dramatically reduces the carbon footprint by 94% and CED by 90%. Among the impact categories at the midpoint level, air transportation dominates fossil depletion, marine eutrophication, natural land transformation, ozone depletion, and photochemical oxidant formation for the 1-hub in-person scenario. And food preparation, electricity consumption at home, and ICT services contribute to the majority of each impact category for the baseline virtual scenario. Moreover, agricultural land occupation, terrestrial ecotoxicity, and water depletion are dominated by food preparation for both in-person and virtual conferences.”

Reviewer comments, second round review

Reviewer #1 (Remarks to the Author):

I appreciate the time and effort the authors have put to address the reviewer comments and applaud deposition of all the data used for the analysis in Github. The authors have addressed all my comments in a satisfactory manner. For the benefit of researchers who will read and assess this valuable work and conference organizers who should be encouraged to assess the parameters and organization of their meeting, I would like to ask the authors to deposit their full commented code to Github as well.

Reviewer #2 (Remarks to the Author):

The authors did a very good job in revising their paper. They have addressed all my remarks and answered all my questions in a very comprehensive way. Particularly the now included sensitivity analyses improve the manuscript in my view a lot.

I think the paper is now clearly ready for being published.

The topic is relevant, the analyses and calculations are well documented and reproducible and the presentation of the results is well accessible for a wide public. From my point of view the manuscript can be published as it stands.

best regards,
Sebastian Jäckle